# Performance of open-path lasers and FTIR spectroscopic systems in agriculture emissions research

Mei Bai[1], Zoe Loh[2], David W. T. Griffith[3], Debra Turner[1], Richard Eckard[1], Robert Edis[1], Owen T. Denmead[4], Glenn W. Bryant[3], Clare Paton-Walsh[3], Matthew Tonini[3], Sean M. McGinn[5], Deli Chen[1]

[1]Faculty of Veterinary and Agricultural Sciences, The University of Melbourne, Parkville, VIC 3010, Australia
[2]CSIRO Oceans & Atmosphere, PMB 1, Aspendale, VIC 3195, Australia
[3]School of Chemistry &Centre for Atmospheric Chemistry, University of Wollongong
Wollongong, NSW 2522, Australia
[4]Deceased, CSIRO Agriculture and Food, GPO Box 1666, Canberra, ACT 2601, Australia
[5]Agriculture and Agri-Food Canada, Lethbridge, Alberta, Canada

*Correspondence to* Mei Bai (mei.bai@unimelb.edu.au)

**Abstract.** The accumulation of gases into our atmosphere is a growing global concern that requires considerable quantification of the emission rates and mitigate the accumulation of gases in the atmosphere, especially the greenhouse gases (GHG). In agriculture there are many sources of GHG that require attention in order to develop practical mitigation strategies. Measuring these GHG sources often rely on highly technical instrumentation originally designed for applications outside of the emissions research in agriculture. Although the open-path laser (OPL) and open-path Fourier transform infrared (OP-FTIR) spectroscopic techniques are used in agricultural research currently, insight into their contributing error to emissions research has not been the focus of these studies. The objective of this study was to assess the applicability and performance (accuracy and precision) of OPL and OP-FTIR spectroscopic techniques for measuring gas mole fraction from agricultural sources. We measured the mole fractions of trace gases methane ($CH_4$), nitrous oxide ($N_2O$), and ammonia ($NH_3$), downwind of point and area sources with known release rate. The mole fractions measured by OP-FTIR and OPL were also input into models of atmospheric dispersion (WindTrax) allowing the calculation of fluxes. Trace gas release recoveries with Windtrax were examined by comparing the ratio of estimated and known fluxes. The OP-FTIR provided the best performance regarding stability of drift in stable conditions. The $CH_4$ OPL accurately detected the low background (free-air) level of $CH_4$; however, the $NH_3$ OPL was unable to detect the background values < 10 ppbv. The dispersion modelling using WindTrax coupled with open path measurements can be a useful tool to calculate trace gas fluxes from the well-defined source area.

**Keywords**: spectroscopy, open path, trace gas, mole fraction, WindTrax modelling

## 1 Introduction

Globally, agriculture contributes approximately 10−12% of anthropogenic greenhouse gases (GHG) entering the atmosphere in 2010 (Smith et al., 2014). The majority of these emissions come from the livestock sector, which includes methane ($CH_4$) from enteric fermentation in ruminants, direct nitrous oxide ($N_2O$) from animal excreta through the nitrification and denitrification processes, and indirect greenhouse effects due to N leaching, runoff, and atmospheric deposition of ammonia ($NH_3$) vitalization from manure by forming $N_2O$ emissions (called indirect

$N_2O$ emissions). Globally, the indirect $N_2O$ emissions account for one third of the total $N_2O$ emissions from
agricultural sector (de Klein et al., 2006).
Direct field measurements of agricultural GHG emissions are difficult due to its high spatial and temporal variation,
diverse source emissions, and lack of appropriate measurement techniques. Consequently, the Intergovernmental
Panel on Climate Change (IPCC, 2006) and Australia's National Greenhouse Gas Inventory Committee (NIR,
2015) use national emission rates that have been based primarily on extrapolations of laboratory and enclosure
measurements. Such extreme extrapolations are subject to greater uncertainty than would be the situation if farm-
scaled values were used. Meeting international obligations on GHG reporting should ultimately require non-
intrusive emission measurements at an appropriate regional scale. Moreover, development, implementation and
adaptation of mitigation strategies rely on well-developed measurement methodologies.
Although considerable effort is being made to document GHG emissions from land-management practices, the
measurement techniques employed in that endeavour are not ideal. Surface chamber method is typically used to
measure gas fluxes from the soil surface, but substantial numbers of surface chambers are required to reduce the
temporal and spatial variations in gas emissions from large scale source (McGinn, 2006). Mass balance techniques
measured emissions from a source area are based on the total influx and efflux of each gas carried into and out of a
control volume (Denmead, 1995). Original applications of this method required the targeted source area to be bounded
by a "fence" of sampling pipes that extended to the upper limit of the gas plume generated from the source. Influxes
and effluxes were calculated by integrating the horizontal fluxes (the product of wind speeds and gas mole fractions)
across the boundaries (Denmead et al., 1998). The plume generated from an area source is expected to extend up to a
height of at least one-tenth of the upwind fetch. Two technological developments together offer a considerable
simplification and flexibility of this basic mass balance technique. The advent of open-path (OP) gas analysers has
enabled the measurement of average mole fractions over long path lengths, removing the need for sampling tubes,
pumps and multiplexing to a closed-path analyser. In addition, mathematical models of atmospheric dispersion allow
fluxes to be inferred from mole fraction measurements and boundary layer wind statistics. Studies of using these
combined OP and dispersion techniques have been reported extensively, such as dairy farms (Bjorneberg et al., 2009;
Harper et al., 2009; VanderZaag et al., 2014), grazing cattle (Laubach et al., 2016; Tomkins et al., 2011), cattle feedlots
(Bai et al., 2015; Loh et al., 2008; McGinn and Flesch, 2018), boiler production (Harper et al., 2010), storage lagoon
(Bühler et al., 2020; McGinn et al., 2008), animal waste treatment (Bai et al., 2020; Flesch et al., 2011; Flesch et al.,
2012), bush fire (Paton-Walsh et al., 2014), geosequestration from industries (Feitz et al., 2018; Loh et al., 2009), and
urban vehicle emissions (Phillips et al., 2019). Although these combined OP and dispersion techniques have
increasingly gained researchers' attentions as a useful tool in measuring gas emissions from large scale field, such as
insight into the OP sensors contributing error to emissions research has not been the focus of these studies.
The purpose of our study is to evaluate these two techniques for measuring GHG emissions from agricultural lands.
Two OP spectroscopic techniques are used to determine line-averaged mole fractions in the field measurements. The
underlying principles of the method and the accuracy and precision of the broad band OP-Fourier transform infrared
spectrometer (FTIR) and single band OP-laser (OPL) spectrometer are tested at experimental sites using releases of
gases at known rates from a point and area sources. We measured the mole fractions (in air) of $CH_4$, $NH_3$ and $N_2O$
with two spectroscopic techniques when gas was released at a known rate. The mole fractions measured by OP-FTIR
and OPL were also input into models of atmospheric dispersion (WindTrax) allowing the calculation of fluxes. Trace
gas release recoveries with Windtrax were examined by comparing the ratio of calculated and known fluxes. This
study would be the first paper of solely comparing the performance between OP mole fraction sensors and provide
the information as reference for measurement techniques in large-scale gas emission research.
**2 Materials and Methods**
**2.1 Experimental design**
The field measurement campaigns were conducted at three sites (Fig. 1):
*Kyabram, Victoria DPI Irrigated dairy research farm* (36.34°S, 145.06°E, elevation 104 m). This is a well-established
research site ideal for micrometeorological measurements, with flat terrain and an existing suite of instrumentation.
Measurements were set up in two adjacent bays near the existing micrometeorological site. The principal disadvantage
of this site was the considerable variation in background trace gas mole fractions (particularly $CH_4$), due to the high
cattle population in the region.
*University of Wollongong* (34.41°S, 150.88°E, elevation 26 m). The No.3 sports oval at the University of Wollongong
is a flat, grassed area approximately 200-250 m in extent. It is surrounded by trees and not a suitable site for
micrometeorological measurements but was well suited to trial release measurements and early OP-FTIR field tests.
*Commercial beef cattle feedlot*, Victoria (225 km northwest of Melbourne, Australia). This site was used for
comparisons of sensors side-by-side experiments. The farm is flat and well suited to micrometeorological
measurements of $CH_4$ emissions from cattle pens.

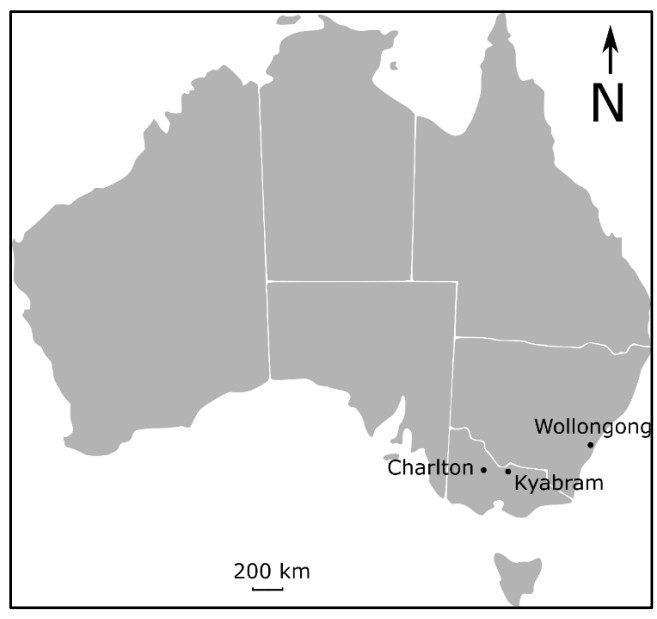

**Figure 1: Three experimental sites at Wollongong sports field, Kyabram research centre, and a feedlot at Charlton.**

The trace gas release measurements including point and area sources were conducted at Kyabram and Wollongong, assuming that all trace gases ($CH_4$, $NH_3$, and $N_2O$) disperse equally from source to open path (OP). Two OP sensors were trialled – a broad band FTIR spectrometer (OP-FTIR) and a single wavelength laser-based instrument (OPL). Besides the gas release measurements, two OP-FTIR sensors were also conducted a side-by-side comparison of measuring gas mole fractions from cattle pens at a commercial beef cattle feedlot. A summary of these trials is shown in Table 1.

**Table 1.   Summary of field measurements at Kyabram, Wollongong, and the Victorian feedlot. Target gases, instrumentations used for the studies, and study durations are also shown.**

| Location | Trial Date | Experiment | Pathlength/m | Height/m | Target Gases | OP sensor[δ] |
|---|---|---|---|---|---|---|
| Kyabram | 25-29 July 2005 | Gas releases, point sources | 137/125 | 0.5 | $CH_4$, $N_2O$, $NH_3$ | OP-FTIR[§] |
| | 1-4 Aug. 2005 | Gas releases, area sources, Side-by-side comparison | 137/125 | Ground | $CH_4$, $NH_3$ | OP-FTIR[§], OPL |
| | 21 Mar. 2006 | Herd of cattle Side-by-side comparison | 227 | 1,7[*] | $CH_4$, $NH_3$ | OP-FTIR, OPL ($CH_4$) |

| | | | | | | |
|---|---|---|---|---|---|---|
| Wollongong | 14-18 May 2005 | Gas releases, point sources, Side-by-side comparison | 87.5/150 | 1.28 | $CH_4$, $N_2O$, $NH_3$ | OP-FTIR[§], |
| | 15-16 Mar. 2006 | Gas releases, point sources, Side-by-side comparison | 148 | 0.5/1.28 | $CH_4$, $NH_3$ | OP-FTIR[§], OPL |
| Commercial feedlot | 28 Feb.- 5 Mar. 2008 | Side-by-side comparison | 100 | 1.7[*] | $CH_4$, $N_2O$, | OP-FTIR[§], OP-FTIR[‡] |

[§] (Bomem)
[‡] (Bruker)
[*] cattle were the main $CH_4$ source, the average of cattle height was 1.7 m.
[δ] the path length for all OP sensors was 1.5 m above the ground.

**Table 2. Gas release times, rates, and source types for controlled release experiments at Kyabram DPI (July-August 2005).**
**Mass flows measured in standard litres per minute (21°C and 1 atm) have been converted to mg s$^{-1}$.**

| | | | OP-FTIR | OPL | Release rates (mg s$^{-1}$) | | |
|---|---|---|---|---|---|---|---|
| Date | Time | Source | Path | Path | $CH_4$ | $NH_3$ | $N_2O$ |
| 27/07/2005 | 10:47 - 12:52 | 1 | 1 | - | 55.37 | 58.80 | 151.95 |
| | 12:52 - 14:17 | 1 | 1 | - | 99.67 | 105.84 | 151.95 |
| | 15:13 - 16:18 | 2 | 1 | - | 99.67 | 105.84 | 151.95 |
| | 17:47 - 08:23 | 2 | 1 | - | 27.69 | 29.40 | 75.97 |
| 28/07/2005 | 10:44 - 14:41 | 2 | 1 | - | 55.37 | 58.80 | 151.95 |
| | 14:41 - 16:42 | 2 | 1 | - | 99.67 | 105.84 | 151.95 |
| | 17:29 - 10:52 | 1 | 1 | - | 27.69 | 29.40 | 75.97 |
| 29/07/2005 | 10:52 - 11:33 | 1 | 1 | - | 11.07 | 11.76 | 30.39 |
| | 11:33 - 12:05 | 1 | 1 | - | 5.54 | 5.88 | 15.19 |
| | 12:43 - 13:51 | 1 | 1 | - | 27.69 | 29.40 | 75.97 |
| | 13:51 - 14:25 | 1 | 1 | - | 55.37 | 58.80 | 151.95 |
| | 14:25 - 15:00 | 1 | 1 | - | 99.67 | 105.84 | 273.51 |
| | 15:00 - 15:30 | 1 | 1 | - | 55.37 | 58.80 | 151.95 |
| | 15:30 - 16:00 | 1 | 1 | - | 11.07 | 11.76 | 30.39 |
| | 16:00 - 16:30 | 1 | 1 | - | 2.77 | 2.94 | 7.60 |
| 1/08/2005 | 15:17 - 15:45 | 1 | 1 | - | 55.37 | 105.84 | 0.00 |
| | 15:45 -16:58 | 1 | 1 | - | 55.37 | 105.84 | 151.95 |

| | | | | | | | |
|---|---|---|---|---|---|---|---|
| | 17:18 - 18:16 | 1 | 1 | - | 55.37 | 0.00 | 303.90 |
| | 18:16 - 09:00 | 3 | 1 | - | 55.37 | 58.80 | 151.95 |
| 2/08/2005 | 12:46 - 16:17 | 3 | 2 | 2‡ | 55.37 | 58.80 | 151.95 |
| | 17:08 - 18:19 | 4 | 2 | 2≠ | 5.54 | 5.88 | 15.19 |
| | 18:19 - 08:55 | 4 | 2 | 2≠ | 5.54 | 0.00 | 15.19 |
| 3/08/2005 | 08:55 - 09:15 | 4 | 2 | 2≠ | 5.54 | 5.88 | 15.19 |
| | 09:15 - 09:33 | 4 | 2 | 2≠ | 0.00 | 2.35 | 0.00 |
| | 09:33 - 10:26 | 4 | 2 | 2≠ | 55.37 | 58.80 | 151.95 |

‡OPL $NH_3$ sensor only. Laser path was located 3 m north of path 2.

≠OPL $NH_3$ and OPL $CH_4$ sensor. OPL $CH_4$ path was located 3 m south of path 2.

## 2.2 Gas release experiments

The underlying principles of the method and the accuracy and precision of the OP-FTIR and laser spectrometers were tested at Kyabram and Wollongong using releases of $CH_4$, $N_2O$, and $NH_3$ at known rates from a common point or area source.

We first conducted the gas release measurements at Kyabram during a period of suitable conditions of steady wind and near neutral stability, and there were no other strong sources of $CH_4$, $N_2O$, and $NH_3$ nearby. Gas release points (sources 1 and 2) were located to the west of the OP-FTIR path 1, which ran N-S along the fence line (Fig. 2). Area sources (sources 3 and 4) were located to the north of the OP-FTIR path 2, which ran NW-SE direction (Fig. 2). The OPL sensors ($NH_3$ and $CH_4$) were set up on the north and south parallel to OP-FTIR path 2, respectively (Fig.2). The path height for all OP sensors was 1.7 m above ground level and the measurement path lengths were 137 and 125 m (two-way path) for paths 1 and 2, respectively. The gas release heights varied from ground level (area sources) to 0.5 m above ground level (point sources). The layout of sources and open path geometries at Kyabram are summarised in Figure 2. A summary of the gas release times, source types and OP sensor measurement paths used at Kyabram is shown in Table 2.

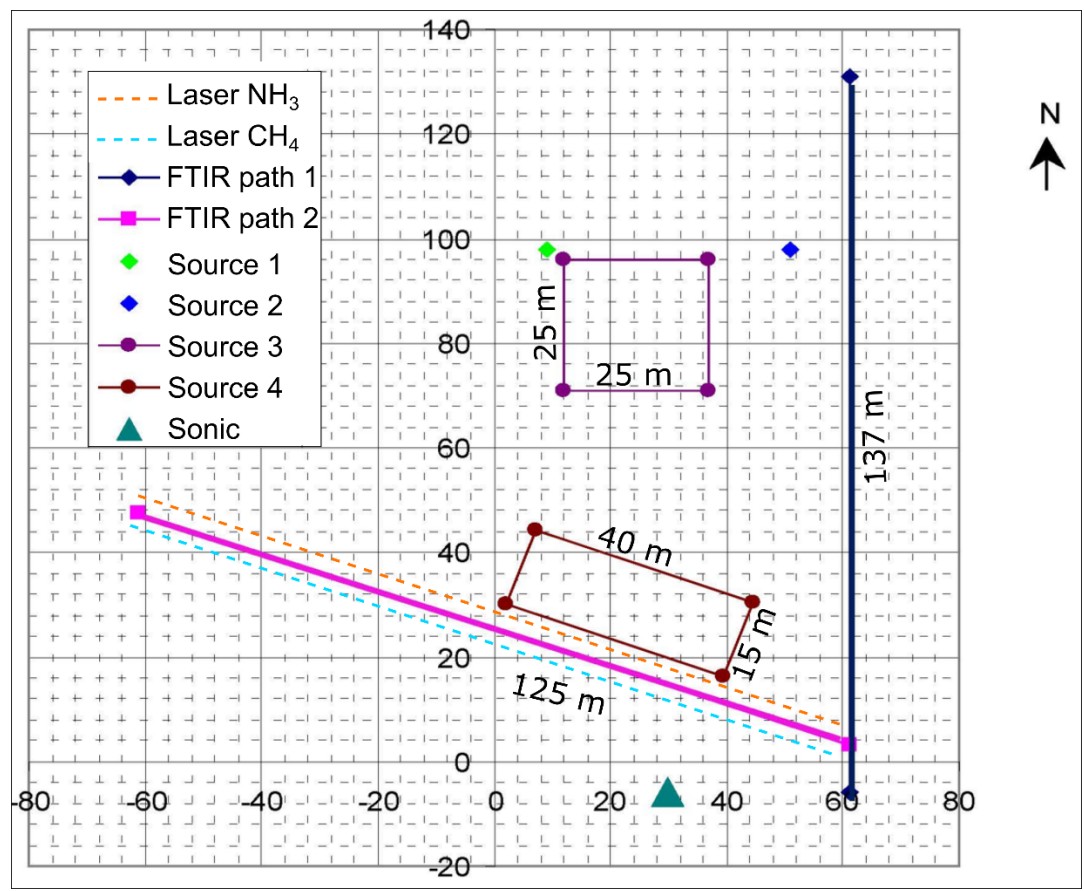


**Figure 2: Point and area gas release sources and OP sensors path geometries (distances in m) at Kyabram July-August**
**2005. Point source 1 is in green and 2 is in blue. Area source 3 is 25 × 25 m, and area source 4 is 40 × 15 m. The OP-FTIR**
**measurement path lengths 1 and 2 were 137 and 125 m (two-way path), respectively. OPL $NH_3$ and $CH_4$ sensor were parallel**
**to OP-FTIR path 2 (dashed yellow and blue lines respectively). Sonic anemometer was located to the south of the site (dark**
**green triangle).**
During the point source release trials, one OP-FTIR was set up on path 1. $CH_4$ and $NH_3$ were released at 9 std L min⁻
¹ (SLPM) and $N_2O$ was released at 5 SLPM, from a single release point, over a three-day study (1-3 August 2005)
(Fig. 2). These were point sources, not distributed as cattle or soil would be. The aim was to show that the known
fluxes can be retrieved from the measurements, for all three gases. In this case it is permissible to have higher emissions
than those typical in the field to minimize uncertainty due to background variability.

The first trial of area source release measurements was undertaken on the evening of 1 August 2005 using the 25 × 25
m area source (source 3) and path 1. Unfortunately, wind conditions were E winds dominated that very little of the
released plume crossed the measurement path. Subsequently, a period in the middle of the day with source 3 and path
2 was employed using the lasers ($NH_3$ only) and one OP-FTIR. The OP-FTIR was set up on the path 2 and laser $NH_3$
sensors were run parallel 3 m north of the OP-FTIR path. Thereafter, the area source 4 (40 × 15 m) and path 2 were
used coupled with the lasers ($NH_3$ and $CH_4$) and the OP-FTIR. Two OPL_$CH_4$ lasers were located 8 m downwind
from the area source, two OPL_$NH_3$ sensors were run parallel 2 m downwind of area source, and OP-FTIR at 5 m
downwind of the source at the same time (Fig. 2). The path height for all OP sensors was 1.7 m and the measurement
path lengths were 137 and 125 m for path 1 and 2, respectively. The different path length was determined depending
on the factors of wind conditions (direction and wind speed) and the distance between the path length and source area.
Given the constant wind direction, the longer pathlength was needed when the measurement path was further away
from the source so that the gas plume could pass by most of the OP measurement path.

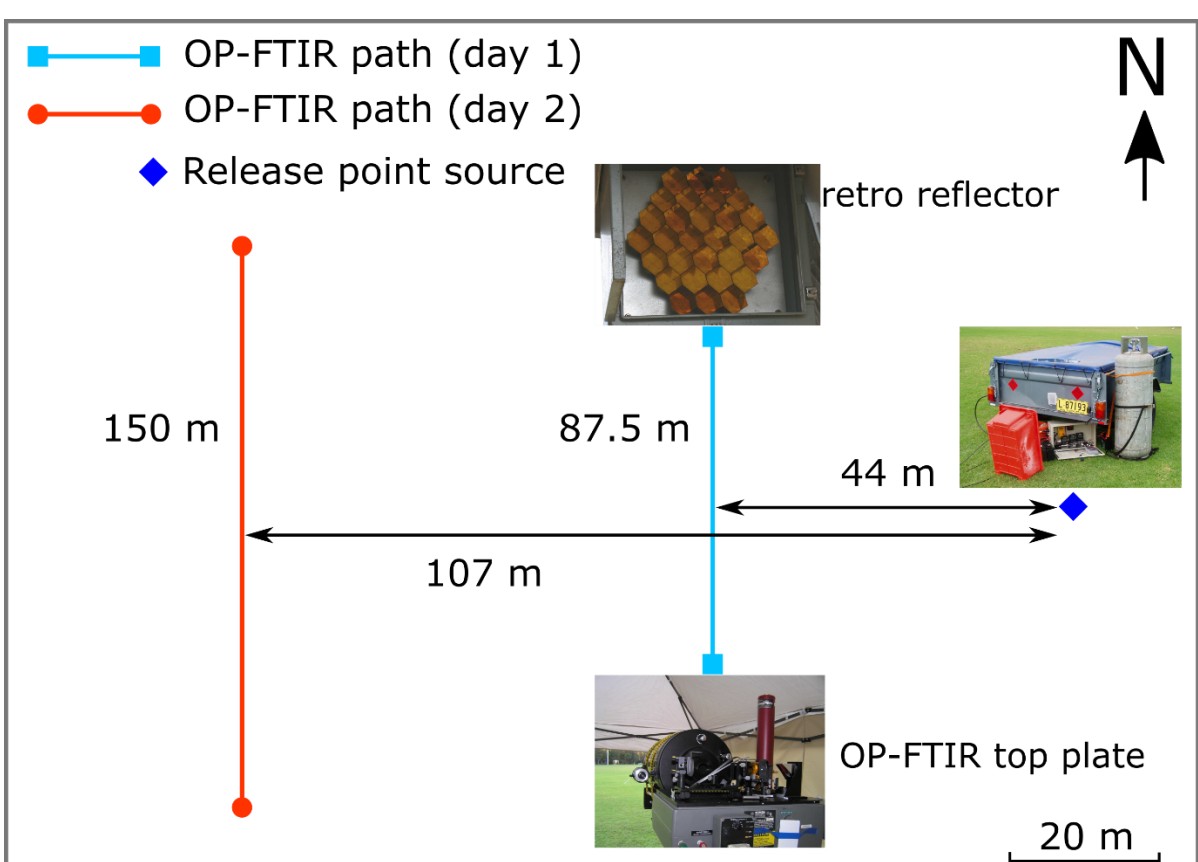

**Figure 3: Point gas release sources and OP-FTIR path geometries (distances in m) at Wollongong August 2005. The OP-**
**FTIR measurement path lengths at day 1 and 2 were 87.5 and 150 m (two-way path), respectively. Three ¼" tubes coming**
**from three tanks (CH₄ (natural gas), NH₃ and N₂O) bundled together on a stake at the release height 1.28 m above ground**
**level.**
The OP-FTIR was also examined at Wollongong sports field during a release trial from for two days (Fig. 3). $NH_3$,
$CH_4$, and $N_2O$ were released at the point source (1.28 m above ground level). The path length of OP-FTIR and its
distance from the source was initially 87.5 (two-way path) and 44 m, respectively, the OP-FTIR was then moved
further away from the source, 107 m from the source with a longer measurement path of 150 m (two-way path).

Furthermore, to check the long-term performance of precision and accuracy of the instruments, we conducted side-
by-side measurements to evaluate sensor differences at Wollongong and a commercial feedlot in northwest of Victoria.
During the intercomparison of OPL and OP-FTIR at Wollongong, the OPL sensors (two for $CH_4$ and two for $NH_3$)
and the Bomem OP-FTIR recorded mole fractions over a path length of 148 m (two-way path) before and after the
gases were released. At the commercial feedlot (Fig. 4), two OP-FTIR spectrometers were run side-by-side. Mole

fractions of $CH_4$, $N_2O$, and $NH_3$ were simultaneously measured for 6 days with the path length of 100 m (two-way path), and measurement height of 1.5 m above the ground. Flasks (600 millilitre, mL) were evacuated prior to gas sampling. Each sample day during stable boundary layer conditions (Monin-Obukov length L, $L \cong 0$–10 m), air samples were collected simultaneously at three points (0, 50, 100 m from the spectrometer) along the measurement path. Total 14 samples over a 5-day period were collected. The air samples were analysed using a closed-path FTIR at the off-site laboratory at University of Wollongong, which has been calibrated to the standard gases $CH_4$ and $N_2O$ (Griffith et al., 2012). The concurrent mole fractions of $CH_4$ and $N_2O$, measured by two FTIR were compared to that of air samples.

**Figure 4: Two OP-FTIR spectrometers (Bomem MB100 and Bruker) during side-by-side operation in a commercial feedlot in Victoria in February 2008.**

**2.3 Gas release system**

The controlled gas releases were of $NH_3$ (>99%, BOC refrigeration grade, Australia), $CH_4$ (compressed natural gas, 89% $CH_4$, Agility, Australia), and $N_2O$ (>99%, BOC Instrument grade, Australia) supplied from high pressure cylinders. Each of the gas flows was controlled by a mass flow controller with ± 2% full scale repeatability (Smart-Trak™ series 100, Sierra Instruments Inc., California, USA). Each gas cylinder was connected to the mass flow controller with 1/4" nylon tubing, the gas outflow from each mass flow controller was released to the atmosphere through another length of nylon tubing. Each gas flow controller was scaled for the gas measurement using the manufacturer's data. Controlled gas flow rates were logged every minute using a data logger (DataTaker, Melbourne). For point-source emissions, the outlets of the three gases were co-located at a release height of 0.5-1.28 m above ground. For surface area emissions, the flows were fed into a length of drip-irrigation tubing (Miniscape, 8 mm) with valve holes every 2.5 m and spread over a $25 \times 25$ m or $40 \times 15$ m grid at ground level.

**2.4 Open-path spectrometers**

**2.4.1 Open-path lasers**

Four open-path lasers (OPL, GasFinder2.0, Boreal Laser Inc, Edmonton, Alberta, Canada) were used. Two units (1012 and 1013) measured $CH_4$, the other two (1015 and 1016) measured $NH_3$. Each OPL was associated with a remote passive retro reflector that delineated the path. The OPL contains a transceiver that houses the laser diode, drive electronics, detector module and micro-computer subsystems. Collimated light emitted from the transceiver traverses the OP to the retro reflector and back. A portion of the beam passes through an internal reference cell. Trace gas mole fraction in the optical path is determined from the ratio of measured external and reference signals. Sample scans are made at approximately 1 s interval and the data were stored internally as one-minute averages. Transceivers are portable, tripod-mounted, battery operated (12 VDC). The retro reflector is tripod-mounted and composed of an array of six gold-coated 6 cm corner cubes with effective diameters of approximately 20 cm. Alignment of transceiver and retro reflector is straightforward and generally stable for several hours over path lengths up to 500 m. The nominal sensitivity of the laser units is 1 part per million-metre (ppm-m), corresponding to 10 ppb for a 100-m path.

**2.4.2 Open-path FTIR**

There were two different OP-FTIR units used in these studies. The first unit consisted of a Bomem MB100-2E OP-FTIR spectrometer (ABB Bomem, Quebec, Canada) and a modified Meade 30.5 cm diameter LX200 Schmidt-Cassegrain telescope that were assembled at the University of Wollongong along with software (Tonini, 2005). Operationally, the transfer optics take the modulated infrared radiation from the FTIR through the telescope to reduce beam divergence to a set of retro reflectors placed at some distance away, collect the returned radiation, and focus the radiation onto a liquid nitrogen cooled MCT detector. A Zener-diode thermometer (type LM335) and a barometer (PTB110, Vaisala, Helsinki, Finland) provide real-time air temperature and pressure data for the analysis of the measured spectra. The spectrometer is operated at $1 \text{ cm}^{-1}$ resolution, and one spectrometer scan takes approximately 4 secs (13 scans $\text{min}^{-1}$). For acceptable signal to noise ratios, scans are generally averaged for at least 1 minute. Immediately following each measurement, the spectrum is analysed (see below) and calculated mole fractions are displayed and logged in real time together with ambient pressure and temperature. Operation is continuous and fully automated by the software to control the spectrometer, data logging and spectrum analysis (Paton-Walsh et al., 2014). Under normal operation the detector must be re-filled with liquid nitrogen once per day, and occasional re-alignment of the spectrometer on the tripod may be required depending on the stability of the tripod footings.

Quantitative analysis to determine trace gas mole fractions from OP-FTIR spectra is based on non-linear least squares fitting of the measured spectra by a computed spectrum based on the HITRAN (high-resolution transmission molecular absorption) database of spectral line parameters (Rothman et al., 2009; Rothman et al., 2005) using a model calculation (Griffith, 1996). The OP-FTIR spectrum is iteratively calculated until a best fit to the measured spectrum is obtained. The mole fraction of absorbing species in the open path is obtained from the best-fit input parameters to the calculated spectrum (Griffith, 1996; Smith et al., 2011). The OP-FTIR spectrometer measures the broadband IR spectrum simultaneously over the range 600-5000 $\text{cm}^{-1}$. The three separate spectral regions ($N_2O$ (2130−2283 $\text{cm}^{-1}$),

CH$_4$ (2920−3020 cm$^{-1}$), and NH$_3$ (900−980 cm$^{-1}$)) are extracted from the broadband spectrum and analyzed separately
for each target species.

The second OP-FTIR unit was the Bruker IRcube spectrometer (Matrix-M IRcube, Bruker Optics, Ettlingen,
Germany) that was developed based on the same principle of Bomem spectrometer (University of Wollongong)
(Paton-Walsh et al., 2014; Phillips et al., 2019). This Bruker OP-FTIR replaced the liquid nitrogen (N$_2$) system by a
Stirling cycle mechanical refrigerator, and a 25.4 cm diameter telescope and a secondary mirror were built to create a
25-mm parallel beam to extend the measurement path up to 500 m. The analytical spectral regions are the same as
Bomem MB 100. More details of Bruker IRcube spectrometer can be found in Bai (2010). The system parameters
from both OP-FTIR are summarized in Table 3. Recently, a custom-made motorised tripod head has been installed to
allow the spectrometer to be aimed at multiple paths where the retro-reflectors were separated vertically or horizontally
(Bai et al., 2016; Flesch et al., 2016).

**Table 3. The system parameters between OP-FTIR Bomem MB100 and Bruker IR cube spectrometer.**

|  | **Bomem MB100** | **Bruker IRcube** |
|---|---|---|
| Detector | Liquid N$_2$ cooled MCT | Stirling cycle refrigerator cooled MCT |
| Size of telescope | 30.5 cm | 25.4 cm |
| SNR[§#] | ~6000 | ~9000 |
| Weight | Heavy | Light |
| Optics dust proof | No | Yes |
| Motorised aiming system | No | Yes |

[§] SNR, signal to noise ratio. A transmission spectrum is calculated by taking ratios of two successive spectra and measuring root
mean square (rms) noise at a spectral region 2500-2600 cm$^{-1}$.
[#] measured over 100 m path length (two-way path).

**2.5. Dispersion modelling (WindTrax)**
To infer emission source strengths or fluxes from atmospheric mole fraction measurements, we require a means to
quantify atmospheric transport and dispersion of the target trace gases between source and measuring point. Our
approach is to infer area-averaged surface fluxes (in excess of background) from measured line-average mole fractions
by using a backward Lagrangian stochastic (bLs) model as developed by Flesch et al. (2004; 1995). The bLs model is
capable of handling sources of arbitrary size and geometry. The model is encoded in the commercially available
software package WindTrax (version 1.0, Thunder Beach Scientific) (Crenna et al., 2006). The inputs for WindTrax
bLs model include the measured mole fraction, sonic anemometer measurements of wind speed and direction, stability
and turbulence as well as other micrometeorological parameters. The WindTrax bLs model simulates the backwards
trajectories of molecules sensed in the optical path.  The instrument tower (in the source area) provides the information
necessary to calculate the trajectories. In this study, 50,000 parcels are released and propagated backward to build up
a statistical distribution of trajectories from which source strengths can be calculated. "Touchdowns" are partitioned
into those originating in the source area and those from the background. This allows the net flux of particles across
the path to be separated into contributions from source and background level.

Similar to the studies in McGinn et al. (2006), we predicted tracer release rates by measuring downwind mole fractions
from area sources using the bLs model. We measured downwind mole fractions before and after releasing each trace
gas, the difference in the mole fractions was then used to determine the source release rate. However, WindTrax cannot
be used to carry out backward simulations for point sources (*i.e.* conversion of mole fraction data to fluxes). It can,
however, predict downwind mole fraction from estimated release rate using the model running in forward mode.
**2.6 Weather data**
A three-dimensional (3-D) sonic anemometer (CSAT3, Campbell Scientific, Logan, Utah, US) with data logger
(CR5000, Campbell Scientific, Logan, Utah, US) were used to record wind speed and direction along with the
turbulence statistics at a frequency of 10 Hz. The fifteen-min interval data were then transformed to friction velocity
($u_*$), atmospheric stability ($L$) and surface roughness length ($z_0$) as half-hour averages, determining the time increments
of OP sensor data.
**2.7 Data filtering criteria**
Poor measurements of mole fractions were not counted when the spectrum signal intensities were < 0.2 (Spec. max)
for OP-FTIR, light level less than 5,000 or greater than 12,000, and $R^2 > 0.90$ for OPL. Following Flesch et al. (2004),
we excluded the data that were associated with error-prone WindTrax fluxes (low wind conditions and strong stable
or unstable stratification): wind speed < 2 m s$^{-1}$, |L| <10, and fraction of "touchdown" < 0.1.
**3 Results and Discussions**
**3.1 OP-FTIR measurements**
The wind was steady from the NNW (325-335°) at 1.8-3.5 m s$^{-1}$ over the measurement period of 14:45-16:30 (local
time) on the 27 July 2005 of Kyabram trial (T1). Between 14:45 and 15:10 and after 16:20 background data were
collected. Figure 5 shows the OP-FTIR measurements of all three gases during this period, expressed as path-average
mole fractions in ppbv after subtracting the background level. We found that the enhanced mole fractions of the source
(downwind minus upwind mole fractions) of $CH_4$, $N_2O$, and $NH_3$ measured by OP-FTIR followed a similar
correspondence.



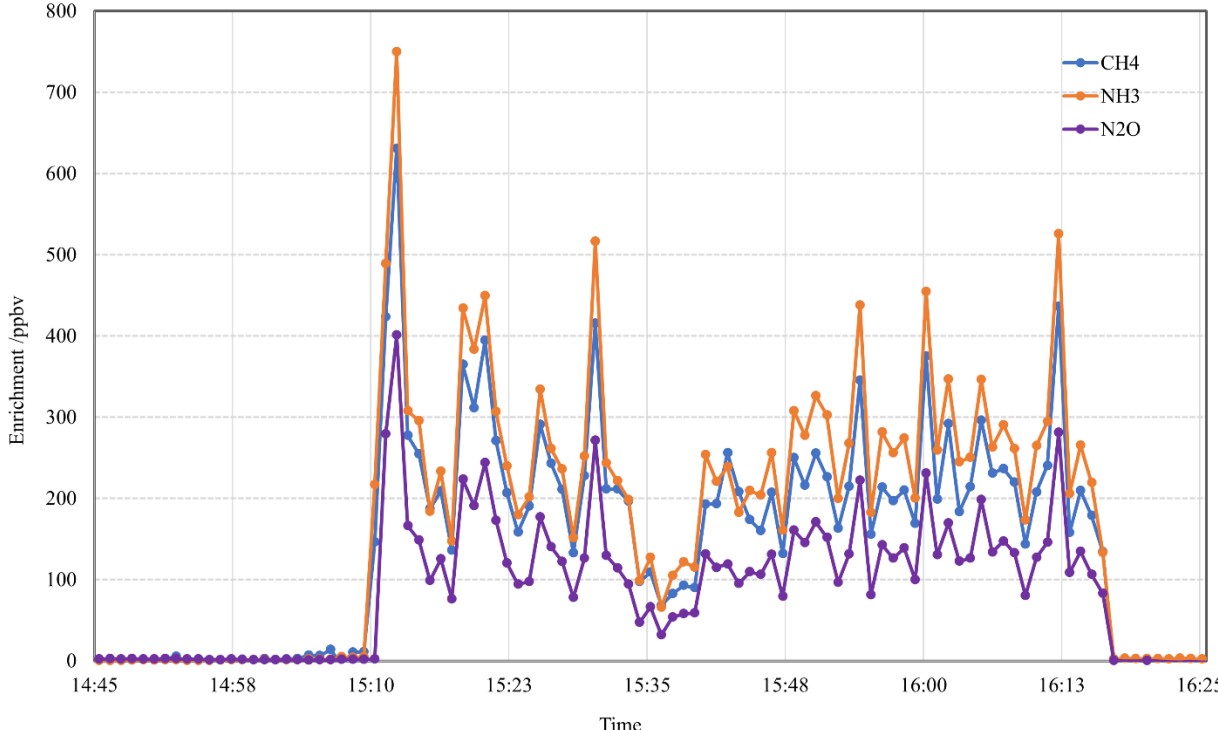


**Figure 5: Measured OP-FTIR one-minute average mole fractions of CH₄, N₂O and NH₃ after subtracting the background levels during a point source gas release experiment at Kyabram on 27 July 2005.**

We also found that the mean measured OP-FTIR mole fraction of $CH_4:N_2O$ was 1.61 compared to the release rate ratio of 1.60, and the mean measured mole fraction of $NH_3:N_2O$ was 1.84 (release rate ratio was 1.80). The release rates with measured regression slopes for all trial release measurements made at both Wollongong and Kyabram are shown in Table 4. In all but three cases the ratio was within 1−8% of the 1:1 ratio. The OP-FTIR system uses no calibration gases but system calibration is based on the accuracy of the HITRAN line parameters and the MALT spectrum model. Typical absolute accuracy is 1−5% depending on species and open path setup, with precision (reproducibility) normally much better than 1% (Esler et al., 2000). The use of MALT synthetic spectra based on quantum mechanical parameters has been shown to yield accurate results (within 5% of true amounts) when tested against calibration gases in a 3.5 L multi-pass gas cell with 24 m optical path length (Smith et al., 2011). In each case of disagreement, the correlation remains strong, and the systematic differences can reasonably be attributed to either a leak in the release system or in the case of low $NH_3$ due to the losses by adsorption at the (wet) ground over the longer release-measurement distance during the experiment.

**Table 4. Comparison of the release rate ratios and OP-FTIR measured enhanced mole fractions for the controlled release gas measurements.**

| Location and time of measurement period[δ] | Distance of gas release (m), height of gas release | Compared gases | Ratio of controlled release rates | Ratio of measured enrichments downwind |
| --- | --- | --- | --- | --- |

| | (m), measurement path distance (m) | | (± 2% measurement error) | (slope of regression ± 95% confidence interval) |
|---|---|---|---|---|
| **Kyabram** | | | | |
| | | | | |
| **(T1)** | | | | |
| Day 1 | 10, 0.5, 137 | $NH_3$, $N_2O$ | $1.800 \pm 0.036$ | $1.841 \pm 0.026$ |
| 1445−1625 h | | $CH_4$, $N_2O$ | $1.602 \pm 0.032$ | $1.609 \pm 0.034$ |
| | | | | |
| Day 2-3 | 10, 0.5, 137 | $NH_3$, $N_2O$ | $1.000 \pm 0.020$ | $1.024 \pm 0.010$ |
| 1730−830 h | | $CH_4$, $N_2O$[&] | $0.890 \pm 0.018$ | $0.946 \pm 0.038$ |
| | | | | |
| Day 2 | 10, 0.5, 137 | $NH_3$, $N_2O$ | $1.000 \pm 0.020$ | $1.028 \pm 0.019$ |
| 900−1440 h | | $CH_4$, $N_2O$ | $0.890 \pm 0.018$ | $0.873 \pm 0.024$ |
| | | | | |
| Day 2 | 10, 0.5, 137 | $NH_3$, $N_2O$ | $1.800 \pm 0.036$ | $1.990 \pm 0.034$[#] |
| 1440−1700 h | | $CH_4$, $N_2O$ | $1.602 \pm 0.032$ | $1.668 \pm 0.049$ |
| | | | | |
| **(T2)** | | | | |
| Day 1 | 52, 0.5, 137 | $NH_3$, $N_2O$ | $1.800 \pm 0.036$ | $1.783 \pm 0.018$ |
| 1545−1625 h | | $CH_4$, $N_2O$ | $0.890 \pm 0.018$ | $0.802 \pm 0.025$[#] |
| | | | | |
| **Wollongong** | | | | |
| | | | | |
| **(T3)** | | | | |
| Day 1 | 44, 1.28, 87.5 | $NH_3$, $N_2O$ | $1.000 \pm 0.020$ | $1.009 \pm 0.020$ |
| 2048−0500 h | | $CH_4$, $N_2O$ | * | * |
| | | | | |
| Day 2 | 107, 1.28, 150 | $NH_3$, $N_2O$ | $1.000 \pm 0.020$ | $0.879 \pm 0.019$[#] |
| 2030−0500 h | | $CH_4$, $N_2O$ | $0.890 \pm 0.018$ | $0.897 \pm 0.032$ |

* no data due to $CH_4$ gas flow problems during this time period.
[#] ratio that is not in agreement with the controlled release ratio ($p < 0.05$).
[δ] time of measurement period represented local time.
[&] the measured mole fraction is from 1730−030 h because of an increased background effect from 030−830 h.
**3.2 OP-FTIR error assessment**
From measurements before and after release, the measurement precision and accuracy of the OP-FTIR measurements
were assessed (Table 5). Measured background mole fractions of $CH_4$ and $N_2O$ at Kyabram were similar to the clean
air values measured at the Cape Grim Baseline Air Pollution Station in Tasmania. The differences between measured
background values at Kyabram and Cape Grim were < 3% and consistent with the known absolute uncertainty in OP-
FTIR calibration (1−5%, the accuracy of MALT and HITRAN).

Regression analyses showed a residual scatter (standard deviation of the residuals) around the regression line of
typically 8 ppbv for $NH_3$:$N_2O$ and 18 ppbv for $CH_4$:$N_2O$ (Fig. 6). This scatter was significantly larger than the
measurement precisions (Table 5) and suggested that the fundamental limit to accuracy and applicability of the OP
technique came from variability in the dispersion of the trace gases by atmospheric turbulence – *i.e.*, even when co-
released at nominally the same point, statistical fluctuations ensured that gas parcels did not follow exactly the same
paths. It thus appeared that measurement precision was not the limiting factor and was sufficient for the purposes of
the measurements. Background variations and turbulence statistics were the error-limiting factors in the OP
measurements.

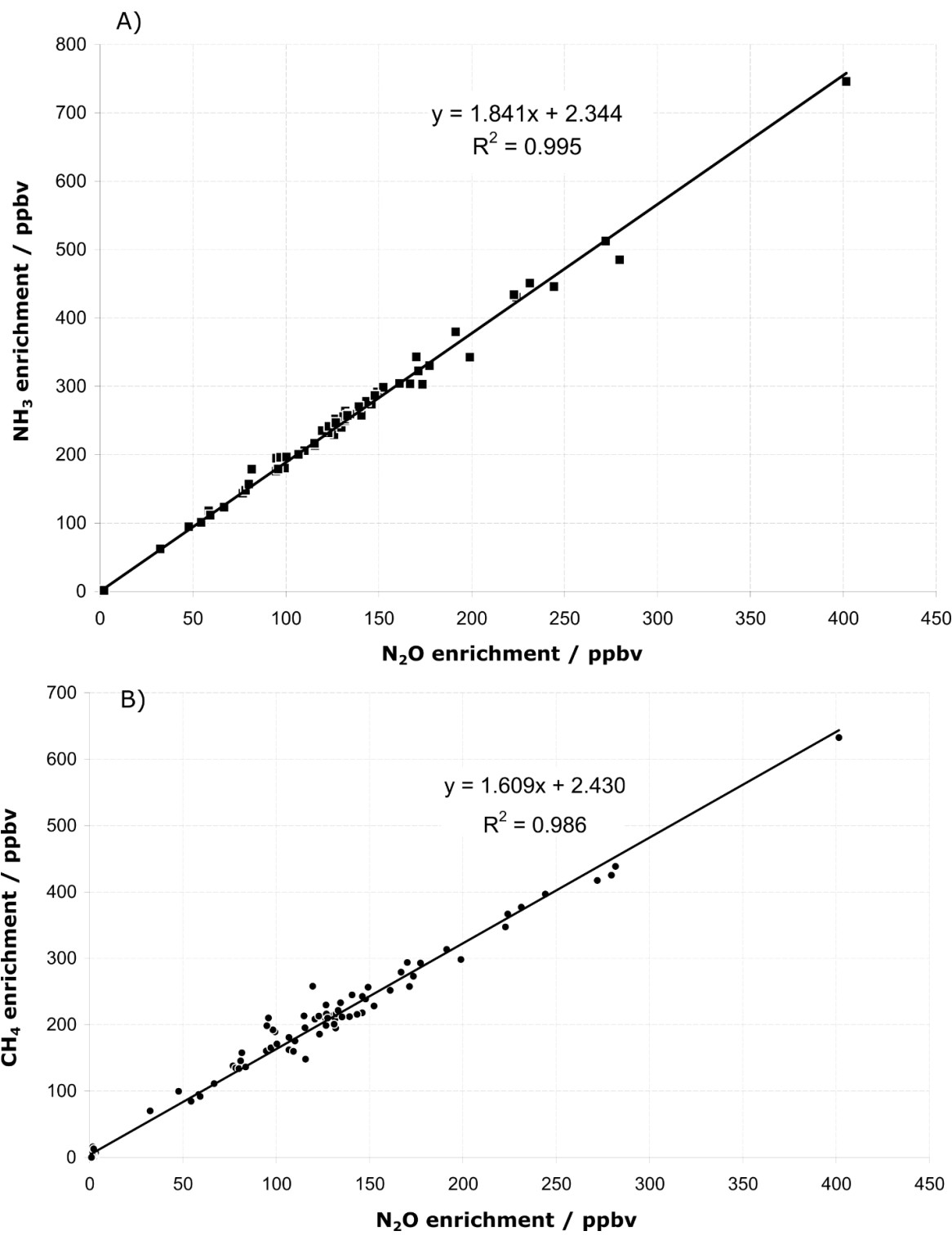


**Figure 6: Regression/correlation analysis of the OP-FTIR measured enrichments shown in Figure 5 between 14:45 and 16:25 of NH₃ vs N₂O (A) and CH₄ vs N₂O (B).**


**Table 5. Measurement precision and comparison with clean air composition for OP-FTIR measurements during the trace**
**gas release trial experimental period at Kyabram. Background mole fractions measured at Cape Grim Baseline Air**
**Pollution Station in Tasmania at the same time are also shown.**

| Target gas | Background measured at Cape Grim | Background measured at Kyabram | Precision typical $1\sigma$ for repeated measurements |
|---|---|---|---|
| $CH_4$ / ppbv | 1738 | 1745 | 3.8 |
| $N_2O$ / ppbv | 317.8 | 310 | 0.3 |
| $NH_3$ / ppbv | 0 | < 1 | 0.4 |

Note:$1\sigma$ is standard error.

## 3.3 Comparisons of OPL and OP-FTIR measurements

The one-minute averages of $CH_4$ and $NH_3$ mole fractions measured by OPL (one unit for $CH_4$, 1012, and two units for $NH_3$, 1015 and 1016) and the OP-FTIR over the period of controlled gas release at Kyabram (T2) were compared (Fig. 7).

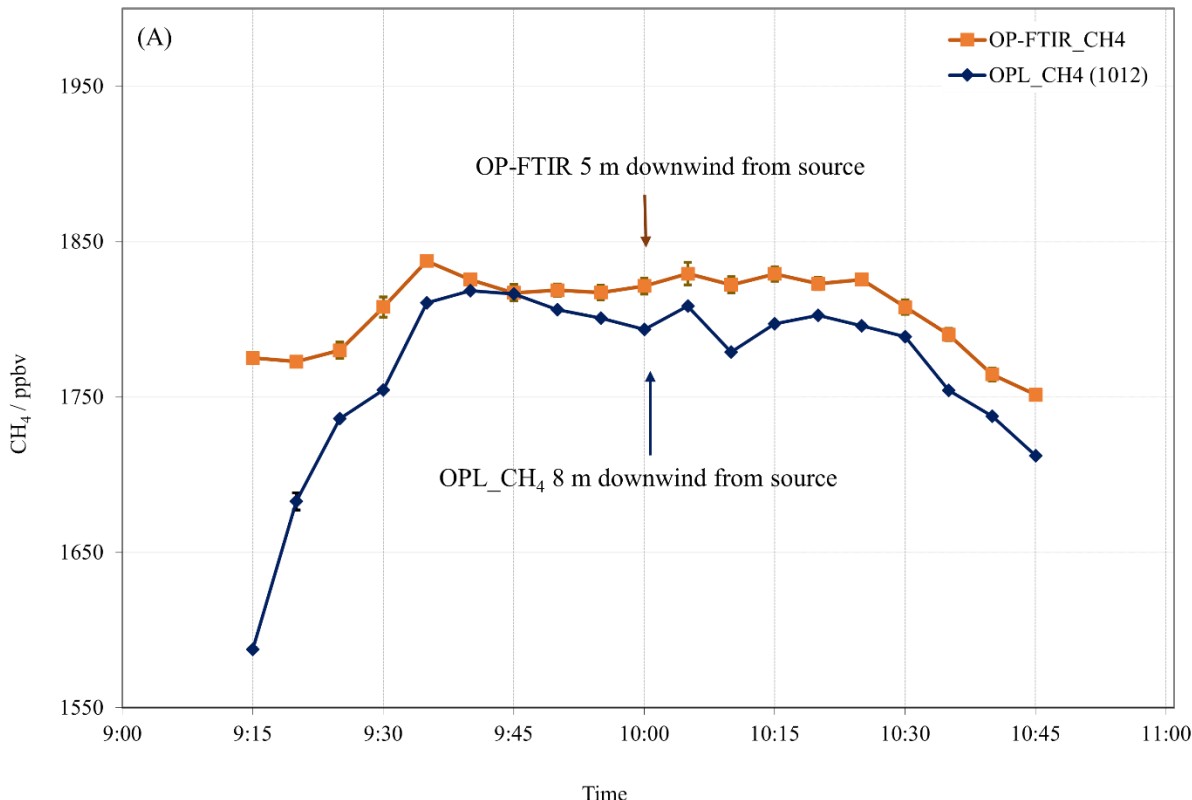

342

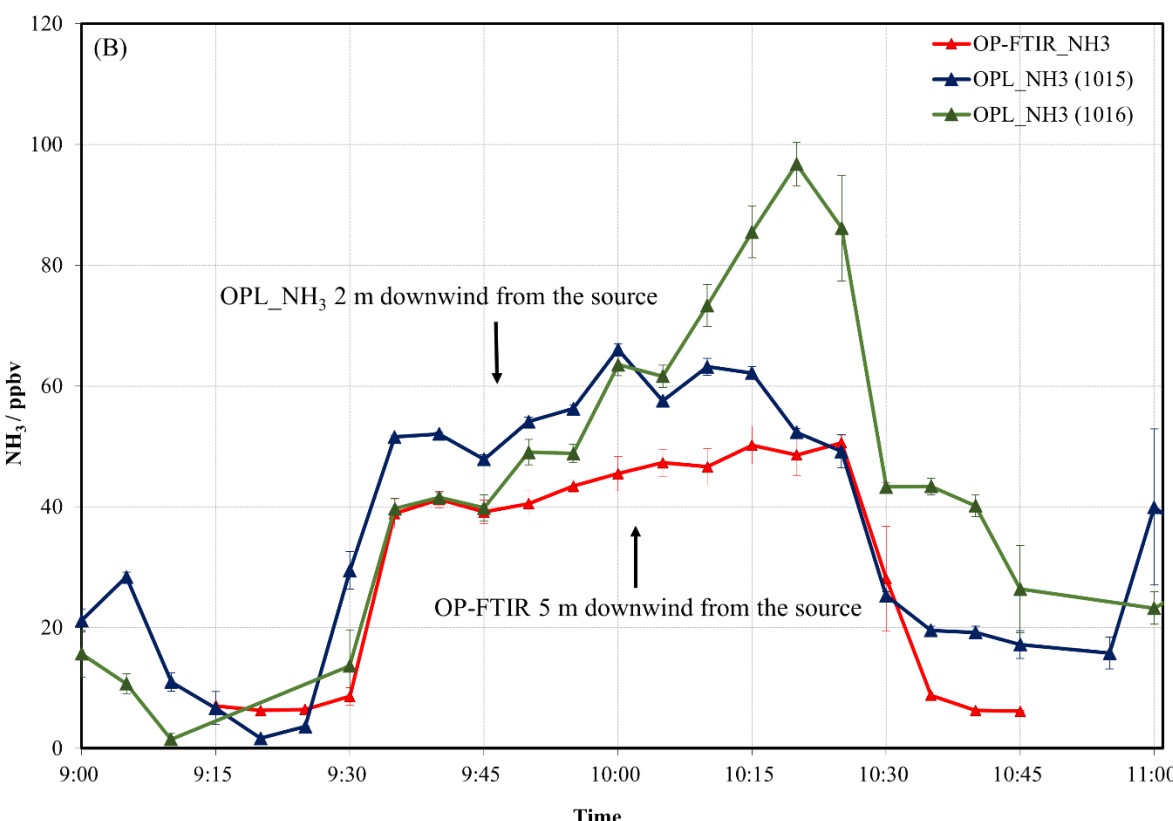

343

**Figure 7: Five-minute averages of $CH_4$ (A) and $NH_3$ (B) mole fraction measurements from the OP-FTIR and OPL downwind of a ground-level grid source 40 × 15 m wide (path length = 125 m) at Kyabram on 3 August 2005 (T2). The error bars represent the standard error.**

In general, the OPL_$CH_4$ and OPL_$NH_3$ tracked the OP-FTIR measurements, however, the OPL_ $NH_3$ did not have a stable baseline (fluctuations of around 15 ppbv) and showed significantly lower signal: noise ratio than that of the OP-FTIR. Offsets in the measured mole fractions may be due to the relative positions of the emission source and the instruments.[1]

A second intercomparison between the $CH_4$ OPL (1012 and 1013) and OP-FTIR measurements at Wollongong is shown in Fig. 8. The thirty-minute averaged OPL_$CH_4$ tracked the OP-FTIR measurements, but recorded lower values, with background $CH_4$ lower than the Cape Grim background of 1738 ppbv (Table 5). There were also discrepancies between the two lasers: 1013 unit was more stable and measured higher values than that of 1012 unit. Flesch et al. (2004) report a similar problem with the long-term stability of $CH_4$ lasers and implement a rigorous calibration strategy, suggesting recalibrating several times over the course of a field campaign. Laubach et al. (2013) reported the temperature-dependent effect on OPL $CH_4$ performance. Implementation of a routine calibration protocol would account for these offsets as long as they were consistent. However, fluctuations of around 10 ppbv characterized the limit on the resolution of the instrument.

---

[1] The laser $CH_4$ mole fractions may be less than those determined by FTIR because the latter's path was only 5 m downwind of the source while the laser path was 8 m downwind. The reverse situation possibly applies to the $NH_3$ measurements, where the $NH_3$ laser path was 3 m upwind of that of the FTIR (Fig. 2).

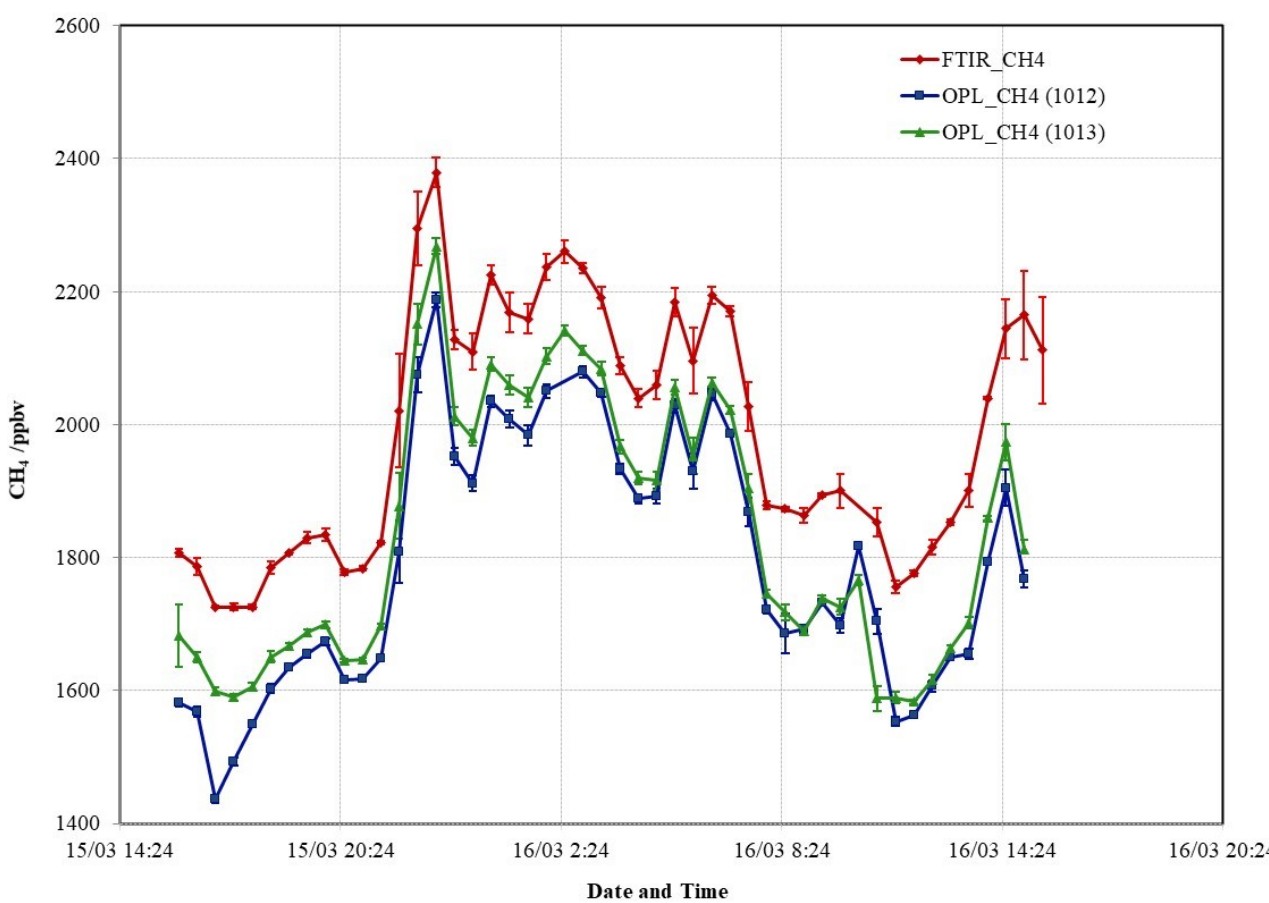


**Figure 8: Thirty-minute averaged CH₄ mole fraction measured by OP-FTIR and both OPL units (1012 and 1013) positioned**
**side-by-side (path length = 148 m) at Wollongong site. Error bars denote the standard error.**



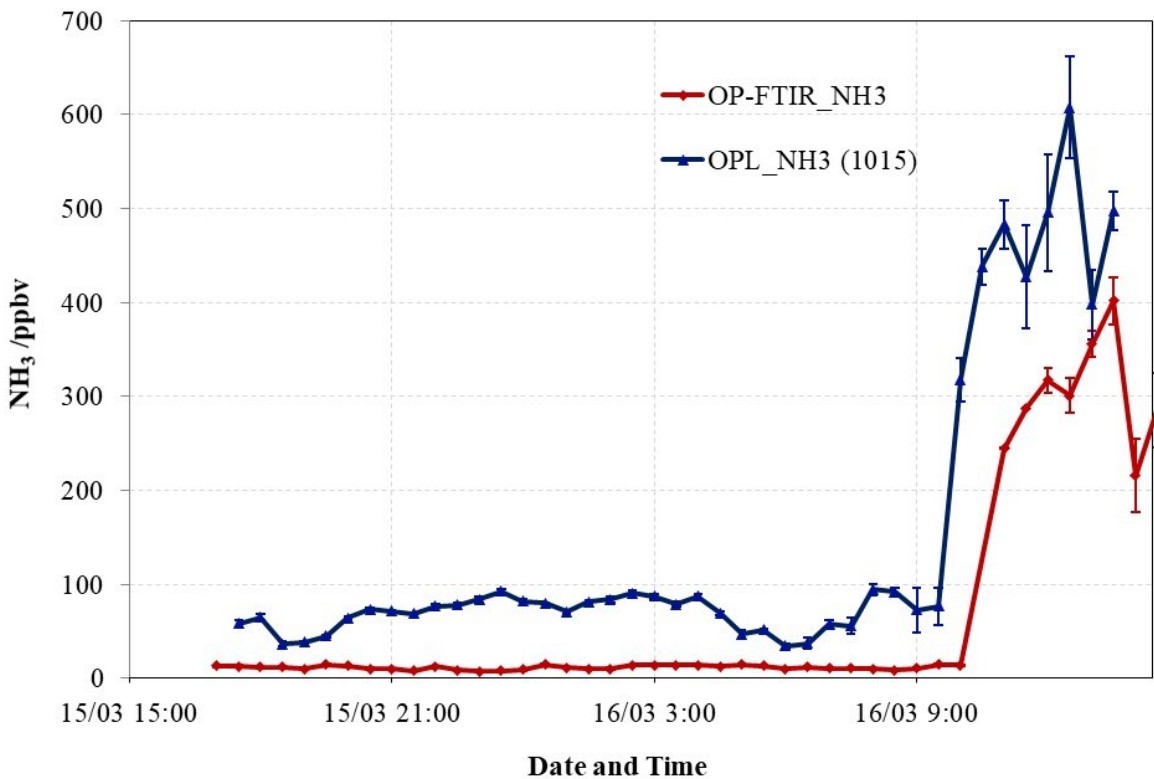


**Figure 9: Thirty-minute averaged NH₃ mole fraction measured by OP-FTIR and OPL unit (1015) positioned side-by-side (path length = 148 m) at Wollongong site. Error bars denote the standard error of the thirty-minute means.**

We also compared thirty-minute averages of $NH_3$ measurements at Wollongong (Fig. 9) prior to and after the gas release ($NH_3$ release rate at 5 L min$^{-1}$). Prior to the gas release (15 March 2006), the laser mole fractions at background levels appear elevated while the FTIR showed greater stable baseline, this suggested clearly that the resolution of the lasers was no better than the 1 ppmv-m specified by the manufacturer. After the $NH_3$ was released (after 10 am 16 March 2006), the path-averaged mole fraction rose above 0.1 ppmv, but the OPL_NH3 (1015 unit) measurements were less erratically at these elevated mole fractions. This indicated the detection limit of the OPL_NH3 was no better than the 1 ppm-m specified by the manufacturer. Rigorous calibration should account for between OPL offsets. However, there remained major discrepancies between measured mole fractions of the OPL_NH3 and OP-FTIR. Clearly, this reflected that the OPL_NH3 are not suited to monitoring background mole fractions of $NH_3$ (typically < 10 ppbv). Moreover, they are only likely to be feasible in situations where there are very large enrichments in $NH_3$ as the precision is no better than 10 ppbv over 100−200 m paths.

**3.4 Comparisons of two OP-FTIR spectrometers**

The ratios of measurement between air samples and FTIR (Bomem and Bruker) are shown in Table 6. We found that $CH_4$ results from Bruker FTIR were more reliable in stable conditions than $N_2O$ values, but comparable in Bomem FTIR results. We also calculated the measurement precisions over a Bruker IRcube which showed higher measurement precision of $CH_4$ and $N_2O$ than Bomem MB100, but similarity in $NH_3$ precision (Table 7).

**Table 6. Ratios of mole fractions of CH$_4$ and N$_2$O between air samples and OP-FTIR including Bomem MB100 and Bruker**
**IRcube spectrometer[#].**

|  | CH$_4$_air/ CH$_4$_FTIR | N$_2$O_air/ N$_2$O_FTIR |
|---|---|---|
| Bomem MB100 | 0.99 (0.03) | 1.01 (0.03) |
| Bruker IRcube | 1.00 (0.03) | 1.04 (0.02) |

[#]mean (standard deviation). The measurements were conducted at stable background conditions for 6 days at Charlton, Victoria.
The pathlength was 100 m (two-way path), and measurement height was 1.5 m above ground level.

**Table 7. The precisions of CH$_4$, N$_2$O, and NH$_3$ for OP-FTIR Bomem MB100 and Bruker IR cube spectrometer.**

|  | *Bomem* | *Bruker* |
|---|---|---|
| Precision[#] |  |  |
| CH$_4$ | 4 ppbv | < 2 ppbv |
| N$_2$O | 0.3 ppbv | < 0.3 ppbv |
| NH$_3$ | 0.4 ppbv | 0.4 ppbv |

[#] measured over 100 m path length (two-way path).

**3.5 Trace gas recoveries with Windtrax**

**3.5.1 OP-FTIR**
We ran Windtrax bLs model to calculate trace gas fluxes during a period in the middle of the day on 2 August 2005
with source 3 (25 × 25 m) and path 1 with the mole fraction measured by the OPL (NH$_3$ only) and OP-FTIR (Fig. 2).
Meteorological conditions varied significantly throughout the period, from unstable (L ≅ -10 m) at the start to slightly
stable (L ≅ 50 m) towards the end. Wind speed averaged 2.5 m s$^{-1}$ and direction was relatively constant at 30°. We
assumed the background mole fraction was constant, 1755, 324, and < 1 ppbv for CH$_4$, N$_2$O, and NH$_3$, respectively
(Table 5). The results of the Windtrax bLs recovery of flux using OP-FTIR mole fractions are illustrated in Appendix
Figure A1 as the ratio of calculated ($Q_{bLS}$) to known (Q) flux. Recoveries of N$_2$O flux were generally good, although
low (average recovery is 0.93). This may be due to an issue with the operation of the grid source (such as the
distribution of gas). NH$_3$ recovery was even lower (mean of 0.71). In this case the adsorption of NH$_3$ on to the grass
may also contributed to a reduction in measured mole fraction (Tonini 2005). Apart from the first thirty-minute period,
which appeared to have been affected an elevated background mole fraction, CH$_4$ flux recoveries were much lower
(mean of 0.52) than for the other gases.

We also calculated trace gas fluxes with area source 4 (40 × 15 m) and path 2 (Fig. 2). Low release rates were
employed, until the final hour when they were increased by an order of magnitude. Meteorological stability was quite
high at the start of the period (L ≅ 0-10 m), gradually becoming less stable during the night and into morning. Wind
speed was correspondingly low (1.5 m s$^{-1}$) at the start and increased to 4 m s$^{-1}$ by the end of the period and wind
direction swung from ENE to NNE. The results for $N_2O$ are shown in Appendix Figure A2 and for $CH_4$ and $NH_3$ in
Appendix Figure A3. The results for the $N_2O$ fluxes were very encouraging. There were some intervals where
retrievals were greater than 1 at the start of the period, and 2 towards the end. The latter occurred at a time when wind
speed increased, and conditions swung from neutral to unstable. Excluding these intervals provided an average ratio
of 1.04, with a standard deviation of 0.15. Few points were available for $NH_3$ as it was not released during the night.
The average for the last two data points was 0.96. Once again, $CH_4$ retrievals were problematic due to variations in
the background mole fraction. With this source geometry and wind field a change in flux of 1 mmole $s^{-1}$ result in a
path averaged change in mole fraction of 50 ppbv. Small variations in the background thus translated to large mass
flux changes (*e.g.*, 1 ppb corresponds to 1/50 mmole $s^{-1}$ = 0.32 mg $s^{-1}$, or 5.8 % of the released flux of 5.5 mg $s^{-1}$).
Under these conditions accurate flux calculation requires a well-defined background mole fraction measurement.
**3.5.2 Lasers ($NH_3$)**
Figure A4 showed the results of the same controlled release experiment described in the OP-FTIR section above.
Again, the bLs model was used to predict the $NH_3$ emission source strength based on OPL $NH_3$ line-averaged mole
fraction measurements.

Although the correlation was reasonable, unlike the recoveries calculated from the OP-FTIR data, the ratio of predicted
to known source strength was greater than 1 for these data. This was not altogether surprising given the consistently
inflated $NH_3$ mole fractions measured by the OPL sensors.
**3.6 Herd emissions using OP-FTIR, OPL and WindTrax**
The study was conducted at Kyabram DPI on March 21, 2006 (Appendix Figure 5A). Appendix Figures A6 and A7
showed the fluxes of $CH_4$ and $NH_3$ due to a herd of 353 dairy cows grazing at Kyabram DPI on March 21, 2006,
calculated using bLs model in WindTrax and OP-FTIR and OPL (for $CH_4$) measured mole fractions. The calculated
$CH_4$ source was variable because the cows were wandering around the paddock (Fig. A6). Clearly marked at the time
when the cows departed the bay (Bay 8) for milking. The $CH_4$ source strength disappeared after this time, as it should.
Missing data points corresponded to periods of time when the average wind speed was less than 2 m $s^{-1}$, when the bLs
model was likely unreliable. The average calculated source strength, based on the OP-FTIR data, was 57.5 µg $m^{-2}$ $s^{-1}$,
equivalent to 292 g $cow^{-1}$ $day^{-1}$. This calculation assumed a uniform background mole fraction of $CH_4$ of 1610 ppbv.
Fluxes based on the upwind and downwind OPL data were strongly correlated with the OP-FTIR results and predicted
an average flux of 48.5 µg $m^{-2}$ $s^{-1}$. The lower value probably reflected the offsets between the instruments. Atmospheric
conditions of the following day were too still to reliably use the data acquired on the second day of grazing. Figure
A7 showed that the OP-FTIR $NH_3$ fluxes ranged from 0.3-0.8 µg $m^{-2}$ $s^{-1}$, with average flux around 0.5 µg $m^{-2}$ $s^{-1}$,
equivalent to 0.7 gN $cow^{-1}$ $day^{-1}$ assuming $NH_3$ volatilisations only occurred during the daytime (8 hours). This was
similar to the $NH_3$ emission fluxes of 0.25 to 2.5 g $cow^{-1}$ $day^{-1}$, measured at the same site and same season (early
April) in 2004 using the combination of passive $NH_3$ sampler and WindTrax (Denmead et al., 2020).
**3.7 WindTrax sensitivity**
A model sensitivity study was undertaken in order to understand how the source strength predicted by WindTrax alters
with variations in a range of input parameters. No sonic anemometer data was used – instead we used simple wind
speed and direction and constructed a surface layer model from local weather conditions and estimates of surface
roughness. Example data from FTIR measurements in Kyabram on 21 March 2006 was used and five input parameters
were varied around the standard conditions. Table 8 below showed how the calculated source strength of $CH_4$ from
the paddock of cows varied with changes in the wind speed, stability, surface roughness, height of sensor and
temperature assumed by the WindTrax model.
**Table 8. Variations in input parameters to WindTrax.**

| Wind Speed | $1.00$ m s$^{-1}$ | $2.00$ m s$^{-1}$ | **$2.67$ m s$^{-1}$** | $3.00$ m s$^{-1}$ | $4.00$ m s$^{-1}$ |
|---|---|---|---|---|---|
| Source strength ($\mu$g m$^{-2}$ s$^{-1}$) | $32 \pm 4$ | $64 \pm 9$ | **$85 \pm 11$** | $96 \pm 13$ | $128 \pm 17$ |


| Stability | **Bright sunshine** | Moderate sunshine | Slight sunshine | Overcast | night < 3/8 cloud | night > 4/8 cloud |
|---|---|---|---|---|---|---|
| Source strength ($\mu$g m$^{-2}$ s$^{-1}$) | **$85 \pm 11$** | $74 \pm 10$ | $74 \pm 10$ | $74 \pm 10$ | $74 \pm 10$ | $74 \pm 10$ |


| Surface Roughness | 2.3 cm | 5 cm | **10 cm** | 12 cm | 15 cm |
|---|---|---|---|---|---|
| Source strength ($\mu$g m$^{-2}$ s$^{-1}$) | $64 \pm 9$ | $64 \pm 8$ | **$85 \pm 11$** | $85 \pm 11$ | $85 \pm 11$ |


| Height of Sensor | 1.4 m | **1.5 m** | 1.6 m | 1.8 cm |
|---|---|---|---|---|
| Source strength ($\mu$g m$^{-2}$ s$^{-1}$) | $88 \pm 8$ | **$85 \pm 11$** | $81 \pm 12$ | $78 \pm 7$ |


| Temperature | 15°C | 20°C | **22°C** | 24°C | 30°C |
|---|---|---|---|---|---|
| Source strength ($\mu$g m$^{-2}$ s$^{-1}$) | $87 \pm 12$ | $86 \pm 11$ | **$85 \pm 11$** | $85 \pm 11$ | $83 \pm 11$ |


The model appeared to be quite robust with respect to height of the sensor, temperature and stability conditions while
changing the assumed surface roughness from 5 to 10 cm altered the predicted fluxes quite markedly. The modelled
source strength scaled with wind speed so accurate meteorological data was a requirement of this technique. It should
also be noted that the Windtrax model was not expected to work well when wind speed was below 2 m s$^{-1}$.
**3.8 The total uncertainty budget**
We want to compute the total uncertainty associated with the difference in mole fraction between upwind and
downwind. There are three uncertainty sources: instrument precision uncertainty, fitting uncertainty, and absorption
cross-section (HITRAN) uncertainty (the latter two are fractional uncertainties and were taken from Paton-Walsh et
al.(2014)) (Table 9). The measurement precision is in units of ppbv and so the fractional uncertainty that this represents
will change with the trace gas mole fraction. The instrument precision uncertainty ($\delta$) associated with upwind
measurement is 1-$\sigma$, and the uncertainty associated with downwind is also 1-$\sigma$. We assume these errors to be
independent. The instrument precision uncertainty in the difference in mole fraction between upwind and downwind
is thus sqrt((1-$\sigma$)^2 + (1-$\sigma$)^2). We then divide this value by the difference in mole fraction to recover the relative
uncertainty due to instrument precision: sqrt((1-$\sigma$)^2 + (1-$\sigma$)^2) / (CH4$_{downwind}$ − CH4$_{upwind}$). $\Delta$CH$_4$ = CH4$_{downwind}$ −
CH4$_{upwind}$. We then add in quadrature the relative measurement uncertainty due to instrument precision with the fitting
and absorption cross-section uncertainties (also expressed in terms of relative uncertainty). For example, for $CH_4$,
when $\Delta CH_4$ was as low as 20 ppbv, we have a relative uncertainty of 0.28 for the instrument precision, 0.02 for fitting
uncertainty, and 0.05 for absorption cross-section uncertainty. The relative uncertainty propagated across these three
components is: sqrt $(0.283^2 + 0.02^2 + 0.05^2) = 0.288$ or 28.8%. When the $\Delta CH_4$ was increased to 50 ppbv or 100
ppbv, the uncertainty declined dramatically to 12.5 and 7.8%, respectively. However, for $N_2O$ and $NH_3$ the uncertainty
was not limited by the mole fraction enhancement but likely attributed to absorption cross-section uncertainty.
**Table 9. Total uncertainty budget.**

| | $CH_4$ | $N_2O$ | $NH_3$ |
|---|---|---|---|
| Measurement precision (ppbv) | 4 | 0.3 | 0.4 |
| Spectral fitting uncertainty (%) | 2% | 4% | 2% |
| Absorption cross-section uncertainty (%) | 5% | 5% | 5% |
| $\delta(\Delta$ trace gas mole fraction$^{\ddagger})/\Delta$ trace mole fraction (%) | | | |
| $\Delta$ trace gas mole fraction (ppbv) | | | |
| 20 | 28.3% | 2.1% | 2.8% |
| 50 | 11.3% | 0.8% | 1.1% |
| 100 | 5.7% | 0.4% | 0.6% |
| Total uncertainty (%) | | | |
| $\Delta$ trace gas mole fraction (ppbv) | | | |
| 20 | 28.8% | 6.8% | 6.1% |
| 50 | 12.5% | 6.5% | 5.5% |
| 100 | 7.8% | 6.4% | 5.4% |

$^{\ddagger}\Delta$ trace gas mole fraction = (trace gas mole fraction)$_{downwind}$ – (trace gas mole fraction)$_{upwind}$
**4 Conclusions**
We have used OP systems for measuring mole fractions of $CH_4$, $N_2O$ and $NH_3$, and evaluated their performance and
precision. Two OP systems for measuring line-averaged gas mole fractions have been evaluated over path lengths up
to about 200 m.

The OP-FTIR system can measure multiple gases simultaneously with excellent precision, e.g., $CH_4$, 2-4 ppbv, $N_2O$,
0.3 ppbv, and $NH_3$, 0.4 ppbv. As the baseline appears to be very stable, we believe OP-FTIR technique has accuracy
for even small enrichments in GHGs. However, the apparatus remains bulky to set up in a field environment, where
access to main power is often difficult. In contrast, the commercial OPL have the advantage of being readily portable
and battery powered. This study has evaluated OPL for $CH_4$ and $NH_3$. These instruments have somewhat poorer
precision than the OP-FTIR, of around 10 ppbv for $CH_4$ and 15 ppbv for $NH_3$. While the OPL should be capable of
following ambient fluctuations in $CH_4$ gas mole fractions, the resolution of the $NH_3$ OPL was greater than the
background mole fractions of $NH_3$, resulting in large errors when calculating fluxes. WindTrax provided accurate
recoveries of known test gas releases from source area and appears to be well suited to analysis of open path
measurements, under suitable meteorological conditions. These experiments highlighted the importance of having a
robust background mole fraction measurement.

Our studies also suggest that the OP-FTIR and OPL are suitable to measure typical enrichments in $CH_4$ and $NH_3$ from
agriculture and useful in calculating fluxes from a variety of agricultural activities, such as free-ranging cattle and
sheep. We recommend that they are also well-suited to concentrated sources such as feedlots, animal sheds and small
enclosures. The OP-FTIR system should also be suited to emissions of $CH_4$ from rice-growing sources and wastewater
lagoons. The OP-FTIR system provides excellent $NH_3$ precision suitable for measuring paddock-scale emissions from
fertiliser (urea, effluent) applications and dung and urine patches. High detection limit and long-term stability of OP-
FTIR enables to measure small changes in $N_2O$ emissions at large-scale from fertilizer treatment, or dairy pastures.
The OPL $NH_3$ has low resolution of free-air mole fraction, in particular weak sources, where the enhanced values are
low and the error in background is minimized.
**5 Appendices**
Appendix A

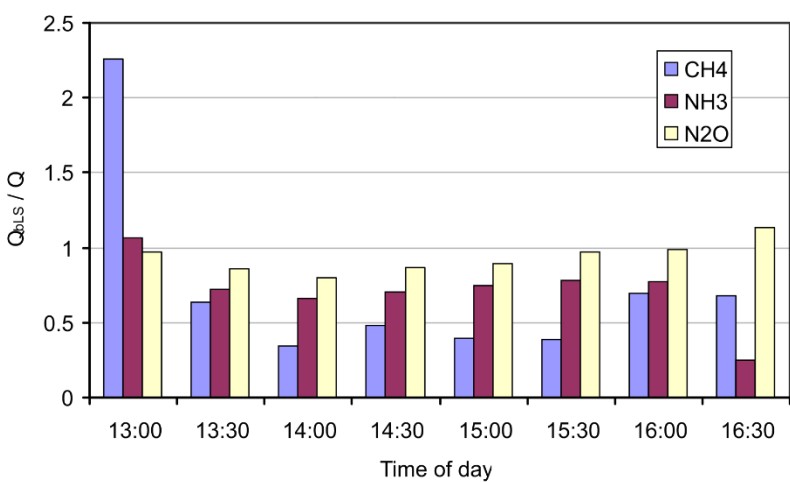

**Figure A1: Ratio of predicted to known flux for ground-level 25 × 25 m area source (Source 3), using OP-FTIR mole**
**fractions and measurement path 2 on 2 August 2005.**

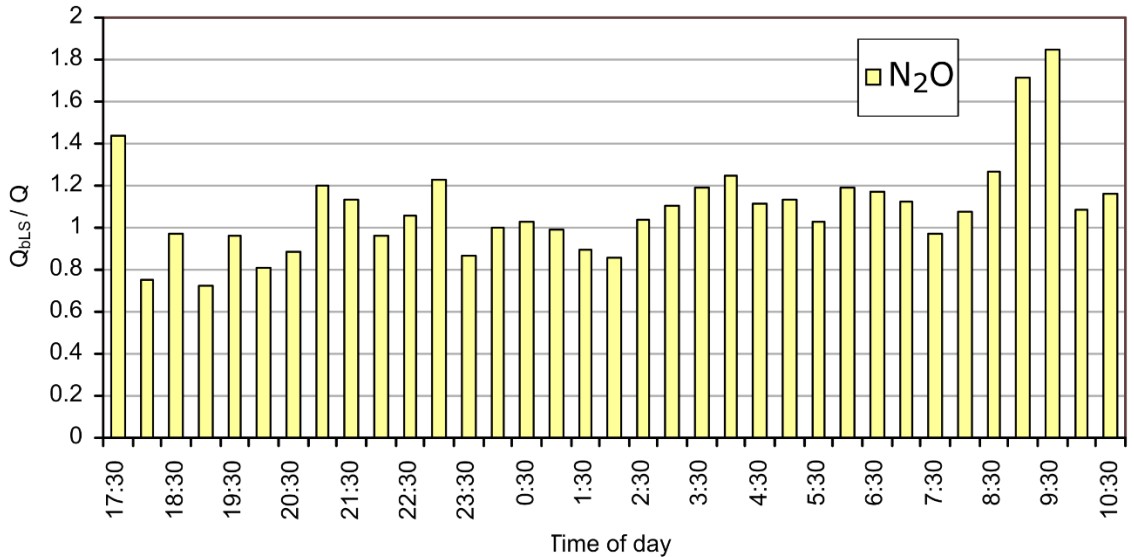

**Figure A2: Ratio of calculated ($Q_{bLS}$) to known N$_2$O (Q) fluxes for the ground-level 40 × 25 m grid source (Source 4),**
**using OP-FTIR mole fractions and measurement path 2.**

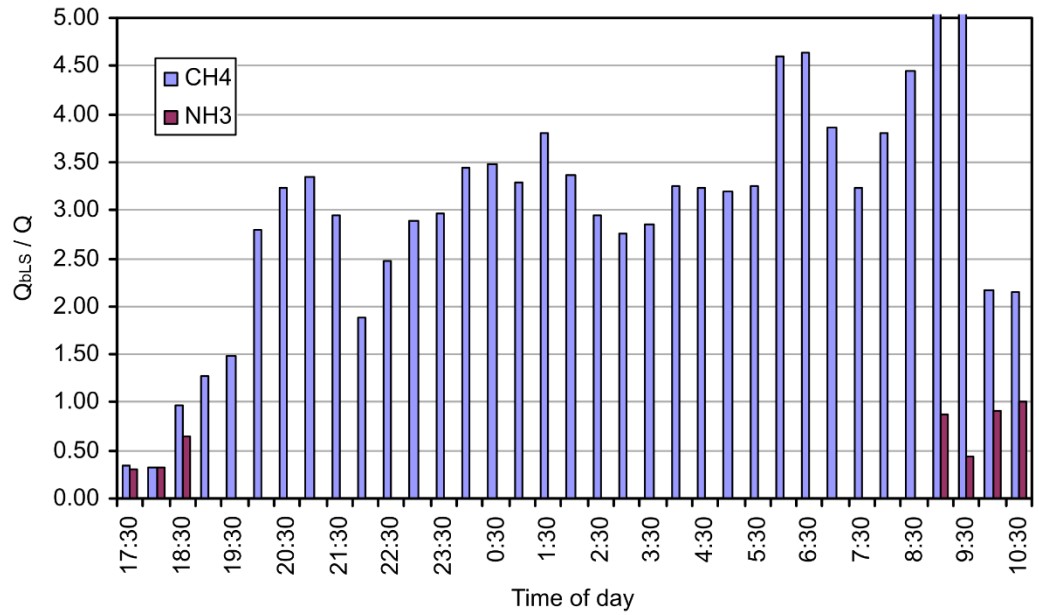

**Figure A3: Ratio of calculated ($Q_{bLS}$) to known CH$_4$ and NH$_3$ (Q) fluxes for the ground-level 40 × 25 m grid source (Source**
**4), using OP-FTIR mole fractions and measurement path 2.**

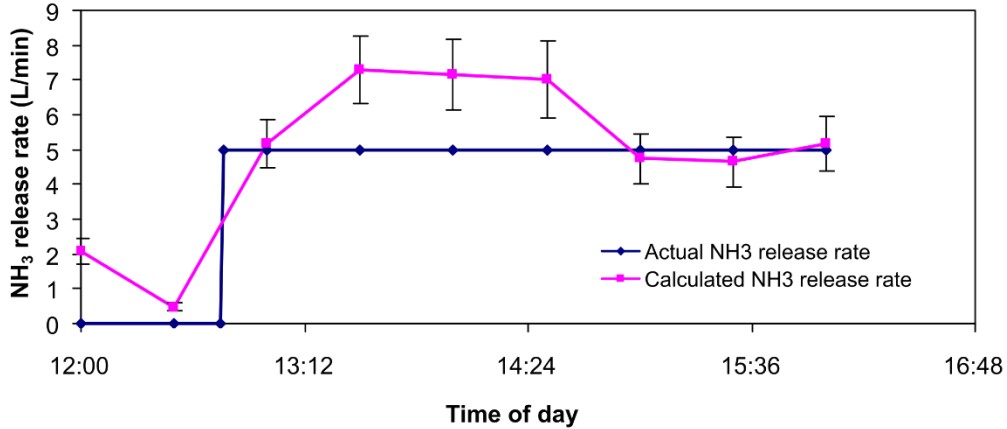

**Figure A4: Controlled release from 25 × 25 m grid (Source 3). Calculated release was average of bLs WindTrax calculations**
**using line-averaged mole fraction measurements from two NH₃ lasers.**

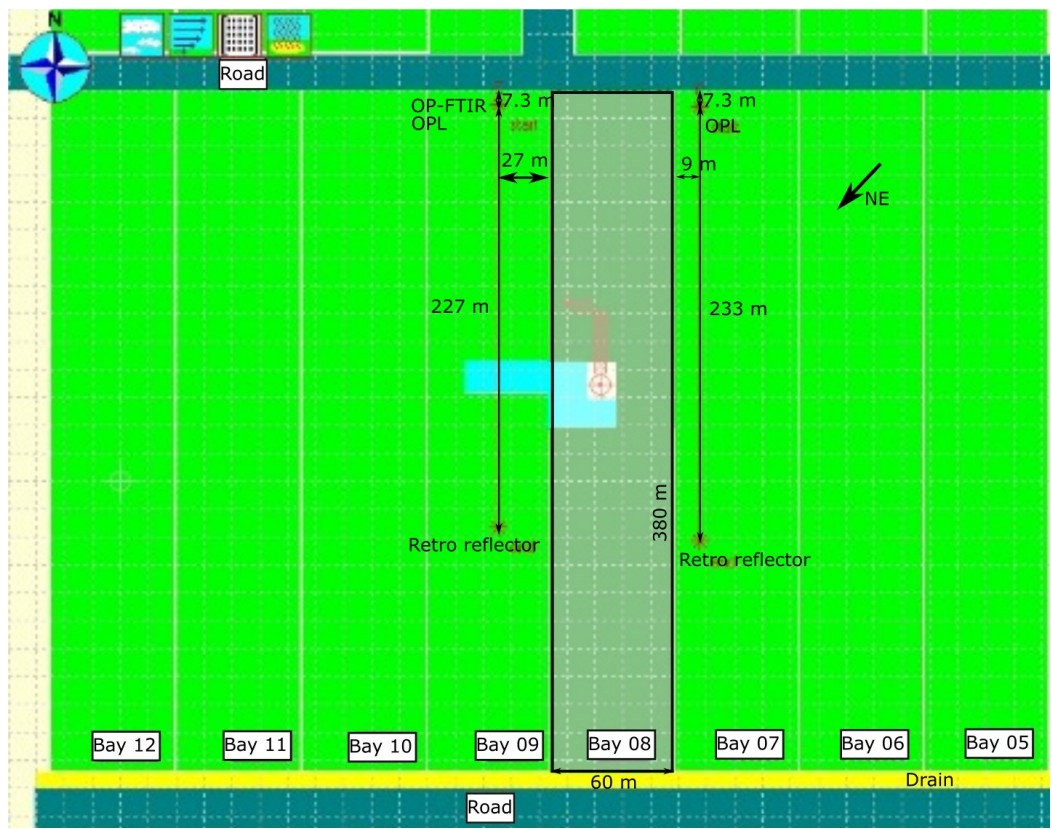

**Figure A5: A WindTrax map showing the layout of herd emissions study at Kyabram on 21 March 2006.**

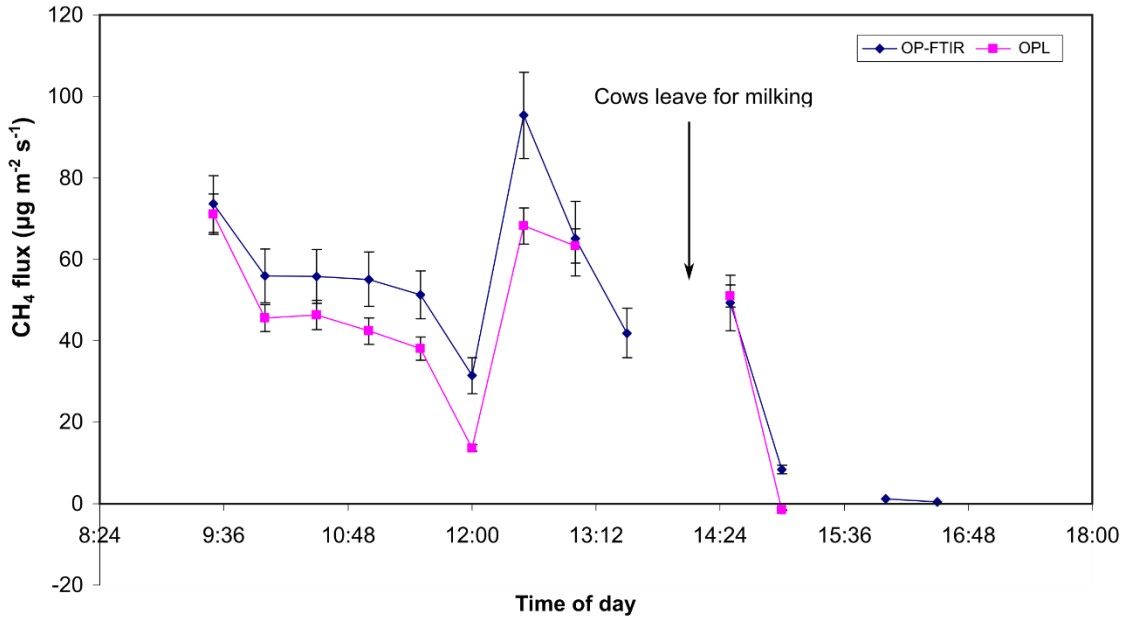

**Figure A6: CH₄ fluxes determined from OP-FTIR and OPL (1012) data and the bLs model at Kyabram 21 March 2006.**

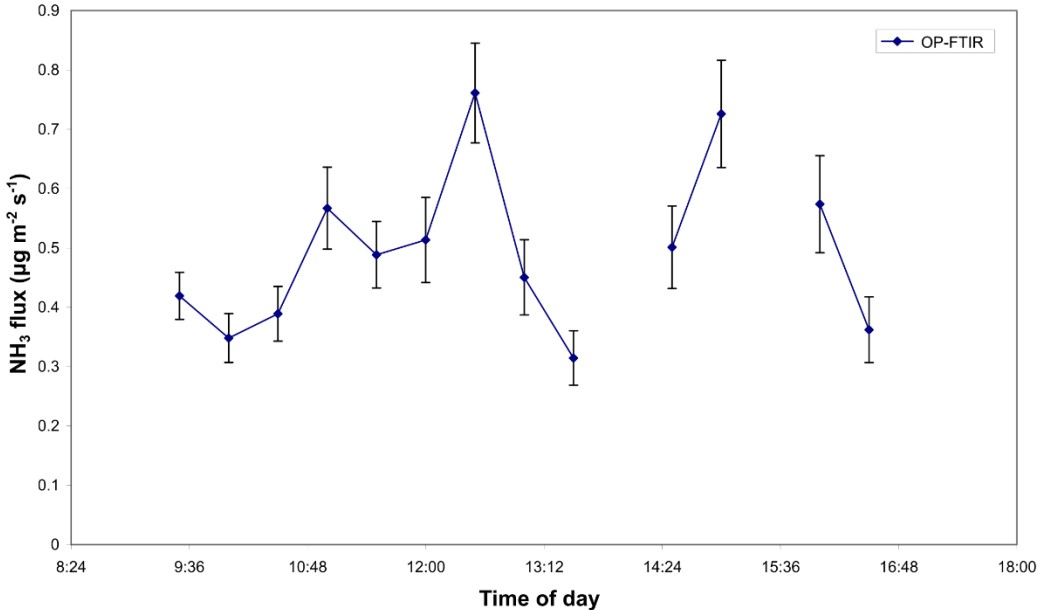

**Figure A7: NH₃ fluxes determined from OP-FTIR data and the bLs model in WindTrax at Kyabram 21 March 2006.**

**6 Data availability**
The raw data are not available to the public. For any inquiry about the data, please contact the corresponding author
(mei.bai@unimelb.edu.au).
**7 Author contributions**
All authors contributed to the conceptualization, methodology, field measurement, data analysis, and draft preparation.

## 8 Acknowledgements

We wish to acknowledge the assistance of many: Ron Teo from the University of Melbourne, the Victorian Kyabram

research station for access to their laboratory and experimental facilities, for provision of micrometeorological data at

Kyabram, and the assistance of their staff, particularly Kevin Kelly, Rob Baigent. We wish to thank also the Australian

Greenhouse Office for their encouragement. Authors would like to thank Travis Naylor, Graham Kettlewell from

University of Wollongong for their assistance during this study.

## 9 Declaration of interests

The authors declare that they have no known competing financial interests or personal relationships that could have

appeared to influence the work reported in this paper.

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
