# Peer review of "Performance of open-path lasers and FTIR spectroscopic systems"

_Atmospheric Measurement Techniques, 2021_

## Author Comment (AC1)

Dear reviewer (Anonymous Referee #1),

Re: Revision of manuscript Number: **amt-2021-347**, Title: **Performance of open-path lasers and FTIR spectroscopic systems in agriculture emissions research**

We thank your positive feedback on the manuscript. We have addressed the comments thoroughly, our response to every issue raised is given point by point in blue text below.

Major comments

The introduction lacks the typical emission ratios of CH4, N2O, NH3, etc, in the agricultural fields. The authors use several cylinders to release these gases at known fluxes, but are these fluxes reasonable compared to the real cases?

Emission ratios of $CH_4$, $N_2O$, $NH_3$ from agricultural fields can be varied. The release rates were point sources, not distributed as cattle or soil would be. The aim is to show that the known fluxes can be retrieved from the measurements, for all three gases. In this case it is permissible to have higher emissions than those typical in the field to minimise uncertainty due to background variability. We have added this in the revised manuscript on page 7, Line 140-142.

It would be very useful to add one section to use the measured VMR from two OP systems to calculate the emission rates. The potential readers are more interested in the fluxes calculation instead of VMR measurements.

Agree with reviewer's comment. we have added a section of flux calculation using backward Lagrangian stochastic dispersion (bLs) modelling (WindTrax) and compared it to the known release rate (recovery study). Please see the section 3.5, 3.6 herd of cattle emission study and 3.7 Windtrax sensitivity in the revised manuscript, page 21-23, the Figures A1-A7 and Table A1 are attached at Appendices. We also added a sentence in the Abstract "The mole fractions measured by OP-FTIR and OPL were also input into models of atmospheric dispersion (WindTrax) allowing the calculation of fluxes. Trace gas release recoveries with Windtrax were examined by comparing the ratio of estimated and known fluxes."

Minor comments:

Line 31: NH3 is an important atmospheric gas, but it is not GHG
Yes, agree with the reviewer. It is worth to mention in the manuscript that ammonia ($NH_3$) is an indirect greenhouse gas, because $NH_3$ deposition can ultimately increase landscape emissions of nitrous oxide ($N_2O$), a potent greenhouse gas (GHG). These "secondary offsite" emissions, generally referred to as indirect emissions, are an important component of the GHG budget of the agricultural sector. The change can be seen in the revised manuscript, pages 1-2, Lines 35-36.

Line42: "relies on" -> reply on.

Yes, agree with the reviewer. The change has been made to be "reply on", see page 2, Line 46.

Line 99: add information on why there are two pathlengths for some experiments.

Yes, we added more information to explain the different path length. "The different path length was determined depending on the factors of wind conditions (direction and wind speed) and the distance between the path length and source area. Given the constant wind direction, the longer pathlength was needed when the measurement path was further away from the source so that the gas plume could pass by most of the OP measurement path." See page 8, Line 152-155.

Line 107: "and 1 atm pressure" -> and 1 atm.

Yes, agree with the reviewer. "the pressure" has been removed in the revised manuscript. See page 5, Line 113.

Line 120: "the measurement path was 137 and 125 m (two-way path) for path 1 and 2, respectively." - > the measurement pathlengths were 137 and 125 m (two-way path) for paths 1 and 2, respectively.

Yes, agree with the reviewer. The change is made to be "above ground level" in the revised manuscript, page 6, Line 126-128.

Line 136: "wind conditions were such poor". Do you have the wind information? The poor wind means the wind direction is opposite? Or the wind speed is too fast?

Yes, we agree with the reviewer. We added the information of poor wind conditions (E winds dominated) in the revised manuscript, page 7, Line 145.

Line 120. "above ground level". Other places use above the ground, please be consistent throughout the whole manuscript.

Yes, changes have been made in the revised manuscript.

Line 145-146 (The layout of the experiment is not shown here): Please add it too.

Yes, we agree with reviewer's comment. In the revised manuscript, we have added the layout of gas release trial at Wollongong on 3 August 2005 (Fig. 3), as well the layout of two OP-FTIR comparisons at a commercial feedlot in Victoria in February 2008 (Fig. 4). See pages 8, 9.

[Figure]

**Figure 3: Point gas release sources and OP-FTIR path geometries (distances in m) at Wollongong August 2005. The FTIR measurement path lengths at day 1 and 2 were 87.5 and 150 m (two-way path), respectively. Three ¼" tubes coming from three tanks (CH₄ (nature gas), NH₃ and N₂O) bundled together on a stake at the release height 1.28 m above ground level.**

[Figure]

**Figure 4: Two OP-FTIR (Bomem MB100 and Bruker) during side-by-side operation in a commercial feedlot in Victoria in February 2008. Each feedlot pen held approximately 100 beef cattle.**

Line 156. First time mentioning CO2 here. It might be better to add some introduction in Section 1.

As suggested by Reviewer 2, $CO_2$ is not relevant to this study, and we agreed to remove the section that was associated with $CO_2$. Please see the change in the revised manuscript.

Line 210: First time mentioning CO here. Same as CO2, it is good to add more information in Section 1.

Same as above response, we have removed CO section in the revised manuscript.

Line 209-210: the three spectral ranges are recorded simultaneously or individually with specific optical filters?

The FTIR spectrometer measures the broadband IR spectrum simultaneously over the range 600-5000 $cm^{-1}$. The three separate spectral regions ($N_2O$ (2130–2283 $cm^{-1}$), $CH_4$ (2920–3020 $cm^{-1}$), and $NH_3$ (900–980 $cm^{-1}$)) are extracted from the broadband spectrum and analysed separately for each target species. Please see the changes in the revised manuscript pages 10-11, Line 229-232.

Line 234 : "14:45- 16:30" the time is local time or UTC?

The time of measurement period was local time. We added "(local time)" in the revised manuscript, please see page 12, Line 282-283.

Line 235: "From 14:45-15:10" -> Between 14:45 and 15:10.

Yes, agree with the reviewer. We made the change in the revised manuscript, please see page 12, Line 283.

Line 310: (data not shown). Please show it.

We added the figure of comparing $NH_3$ mole fractions between two lasers and OP-FTIR (Fig. 9) on page 20.

[Figure]

**Figure 9: Thirty-minute averaged NH₃ mole fraction measured by OP-FTIR and OPL unit (1015) positioned side-by-side (path length = 148 m) at Wollongong site.**

Line 322: what is "in stable conditions"?

Monin-Obukov length L, L≅ 0–10 m. we added it in the revised manuscript, page 21, Line 420.

Line 323: "Carbon dioxide results from both FTIRs were lower than those of air samples by approximately 15%." Do you have a reason for that?

We have removed this statement as $CO_2$ is out of this study.

---

## Author Comment (AC2)

Dear reviewer (Anonymous Referee #2),

Re: Revision of manuscript Number: **amt-2021-347**, Title: **Performance of open-path lasers and FTIR spectroscopic systems in agriculture emissions research**

We thank your positive feedback on the manuscript. We have addressed the comments thoroughly, our response to every issue raised is given point by point in blue text below.

Reviewer suggested to add a discussion on uncertainty budget and calculate emission rates as a real-case scenarios.

We agree with the reviewer. The initial manuscript described the measurements of side-by-side FTIR-laser comparisons and demonstrated excellent correlations between measured line-averaged mole fractions. Ratios of mole fractions in most cases were equal to ratios of release rates within 2%. The measurements confirmed the underlying principle that the released gases were equally dispersed by the atmosphere between source and measurement path. Scatter in measured ratios was larger than the measurement precision would imply and suggests that the random nature of atmospheric dispersion is the limiting factor to accuracy in the overall open path techniques, not instrumental measurement precision.

In the revised manuscript, following reviewer's suggestion, we added three sections:

1, the trial release experiments at Kyabram in July-August 2005 were also analysed using WindTrax to calculate gas source strengths from measured line average mole fractions for each gas. For uniform area sources, excellent recoveries were consistently obtained for $N_2O$ with Windtrax-calculated mole fractions in agreement with measured mole fractions within 5%. For $CH_4$ the results were often affected by additional local sources of $CH_4$ (local dairy herd) and for $NH_3$ there was evidence of losses at the ground surface leading to reduced $NH_3$ recoveries.

2, We also added a study on measuring GHG emissions from a herd of grazing cattle (353 cattle) using side-by-side OP-FTIR and OPL with sonic anemometer data and WindTrax for dispersion analysis. This experiment assessed the usefulness of the WindTrax approach when the source is not strictly uniformly distributed (grazing cattle for $CH_4$, scattered dung and urine patches for $NH_3$). Realistic results were obtained (average 290 g cow$^{-1}$ day$^{-1}$ for $CH_4$, and 0.3-0.8 µg m$^{-2}$ s$^{-1}$ for $NH_3$), which was comparable to the results from our previous study. The $CH_4$ laser results for this trial are in good agreement with the FTIR results.

3, we added total uncertainty budget:

We want to compute the total uncertainty associated with the difference in mole fraction between upwind and downwind. There are three uncertainty sources: instrument precision uncertainty, fitting uncertainty, and absorption cross-section (HITRAN) uncertainty (the latter two are fractional uncertainties and were taken from Paton-Walsh et al. 2014). The measurement precision is in units of ppbv and so the fractional uncertainty that this represents will change with the trace gas mole fraction. The instrument precision uncertainty

(δ) associated with upwind measurement is 1-σ, and the uncertainty associated with downwind is also 1-σ. We assume these errors to be independent. The instrument precision uncertainty in the difference in mole fraction between upwind and downwind is thus sqrt((1-σ)^2 + (1-σ)^2). We then divide this value by the difference in mole fraction to recover the relative uncertainty due to instrument precision: sqrt((1-σ)^2 + (1-σ)^2) / ($CH4_{downwind}$ − $CH4_{upwind}$). $\Delta CH_4 = CH4_{downwind} − CH4_{upwind}$. We then add in quadrature the relative measurement uncertainty due to instrument precision with the fitting and absorption cross-section uncertainties (also expressed in terms of relative uncertainty). The unit for precision (1-σ) is ppbv, and % for uncertainty.

For example, for $CH_4$, when $\Delta CH_4$ was as low as 20 ppbv, we have a relative uncertainty of 0.28 for the instrument precision, 0.02 for fitting uncertainty, and 0.05 for absorption cross-section uncertainty. The relative uncertainty propagated across these three components is: sqrt (0.283^2 + 0.02^2 + 0.05^2) = 0.288 or 28.8%. In this case, total uncertainty is dominated by the uncertainty due to instrument precision. When the $\Delta CH_4$ was increased to 50 ppbv or 100 ppbv, the uncertainty declined dramatically to 12.5 and 7.8%, respectively. However, for $N_2O$ and $NH_3$ the uncertainty was not limited by the mixing ratio enhancement but likely attributed to absorption cross-section uncertainty.

**Table 9. Total uncertainty budget**

| | | $CH_4$ | $N_2O$ | $NH_3$ |
|---|---|---|---|---|
| Measurement precision (ppbv) | | 4 | 0.3 | 0.4 |
| Spectral fitting uncertainty (%) | | 2% | 4% | 2% |
| Absorption cross-section uncertainty (%) | | 5% | 5% | 5% |
| δ(Δ trace gas mole fraction[‡])/Δ trace mole fraction (%) | | | | |
| | Δ trace gas mole fraction (ppbv) | | | |
| | 20 | 28.3% | 2.1% | 2.8% |
| | 50 | 11.3% | 0.8% | 1.1% |
| | 100 | 5.7% | 0.4% | 0.6% |
| Total uncertainty (%) | | | | |
| | Δ trace gas mole fraction (ppbv) | | | |
| | 20 | 28.8% | 6.8% | 6.1% |
| | 50 | 12.5% | 6.5% | 5.5% |
| | 100 | 7.8% | 6.4% | 5.4% |

[‡]Δ trace gas mole fraction = (trace gas mole fraction)downwind − (trace gas mole fraction)upwind

Minor comments:

1) please show the layout of the experiments at three experimental sights.

Agree with the reviewer. we have added the layout of the experimental site at Wollongong (Fig. 3) and commercial feedlot (Fig. 4).

[Figure]

**Figure 3: Point gas release sources and OP-FTIR path geometries (distances in m) at Wollongong August 2005. The OP-FTIR measurement path lengths at day 1 and 2 were 87.5 and 150 m (two-way path), respectively. Three ¼" tubes coming from three tanks (CH$_4$ (nature gas), NH$_3$ and N$_2$O) bundled together on a stake at the release height 1.28 m above ground level.**

[Figure]

**Figure 4: Two OP-FTIR (Bomem MB100 and Bruker) during side-by-side operation in a commercial feedlot in Victoria in February 2008. Each feedlot pen held approximately 100 beef cattle.**

2) measurement data of CO and CO2 is presented in e.g. 2.4.2, however, it is out of the scope of the paper. Please remove or elaborate more.

Agree. we have removed the section about CO and $CO_2$ in the revised manuscript.

3) L272, not shown data is important for the discussion on the instrument precision, please include in the main text.

Agree. we have added Figure 6 in the revised manuscript.

[Figure]

**Figure 6: Regression/correlation analysis of the OP-FTIR measured enrichments in Figure 5 between 14:45 and 16:25 of NH$_3$ vs N$_2$O (A) and CH$_4$ vs N$_2$O (B).**

4) L310 (data not shown), data can be included in the supplement.
Agree. we have added Figure 9 in the revised manuscript, please see page 20.

[Figure]

**Figure 9: Thirty-minute averaged NH₃ mole fraction measured by OP-FTIR and OPL unit (1015) positioned side-by-side (path length = 148 m) at Wollongong site. Error bars denote the standard error of the thirty-minute means.**

5) Figures 3-5, please include the Y - error bars.

We added standard errors (s.e.)  to the five-minute averages of CH₄ and NH₃ mole fraction measured at Kyabram (Fig. 7) and s.e. to the thirty-minute mean of CH₄ mole fraction measured by OP-FTIR and OPL at Wollongong (Fig. 8). However, we did not add error bars in Figure 5 as each dot represent a single measurement (i.e., raw data) which makes it impossible to compute the standard error associated with each dot.

[Figure]

[Figure]

**Figure 7: Five-minute averages of CH₄ (upper) and NH₃ (lower) mole fraction measurements from the OP-FTIR and OPL downwind of a ground-level grid source 40 × 15 m wide (path length = 125 m) at Kyabram on 3 August 2005 (T2). Error bars represent the standard error.**

[Figure]

**Figure 8: Thirty-minute averages of CH₄ mole fraction measured by OP-FTIR and both OPL units (1012 and 1013) positioned side-by-side (path length = 148 m) at Wollongong site. Error bars denote the standard error.**

---

## Author Response (AR2)

**Title. Performance of open-path lasers and FTIR spectroscopic systems in agriculture emissions research**

**Authors**. Mei[1], Zoe[2], David[3], Debra[1], Richard[1], Robert[1], Owen[4], Glenn[3], Clare[3], Matthew[3], Sean[5], Deli[1]

[1]Faculty of Veterinary and Agricultural Sciences, The University of Melbourne, Parkville, VIC 3010, Australia

[2]CSIRO Oceans & Atmosphere, PMB 1, Aspendale, VIC 3195, Australia

[3]School of Chemistry &Centre for Atmospheric Chemistry, University of Wollongong

Wollongong, NSW 2522, Australia

[4]Deceased, CSIRO Agriculture and Food, GPO Box 1666, Canberra, ACT 2601, Australia

[5]Agriculture and Agri-Food Canada, Lethbridge, Alberta, Canada

*Correspondence to* Mei Bai (mei.bai@unimelb.edu.au)

**Abstract.** The accumulation of gases into our atmosphere is a growing global concern that requires considerable quantification of the emission rates and mitigate the accumulation of gases in the atmosphere, especially the greenhouse gases (GHG). In agriculture there are many sources of GHG that require attention in order to develop practical mitigation strategies. Measuring these GHG sources often rely on highly technical instrumentation originally designed for applications outside of the emissions research in agriculture. Although the open-path laser (OPL) and open-path Fourier transform infrared (OP-FTIR) spectroscopic techniques are used in agricultural research currently, insight into their contributing error to emissions research has not been the focus of these studies. The objective of this study was to assess the applicability and performance (accuracy and precision) of OPL and OP-FTIR spectroscopic techniques for measuring gas mole fraction from agricultural sources. We measured the mole fractions of trace gases methane ($CH_4$), nitrous oxide ($N_2O$), and ammonia ($NH_3$), downwind of point and area sources with known release rate. The mole fractions measured by OP-FTIR and OPL were also input into models of atmospheric dispersion (WindTrax) allowing the calculation of fluxes. Trace gas release recoveries with Windtrax were examined by comparing the ratio of estimated and known fluxes. The OP-FTIR provided the best performance regarding stability of drift in stable conditions. The $CH_4$ OPL accurately detected the low background (free-air) level of $CH_4$; however, the $NH_3$ OPL was unable to detect the background values < 10 ppbv. The dispersion modelling using WindTrax coupled with open path measurements can be a useful tool to calculate trace gas fluxes from the well-defined source area.

Re: Revision of manuscript Number: **amt-2021-347**, we thank all the reviewers' valuable feedback on the manuscript. We have addressed the comments thoroughly, our response to every issue raised is given point by point in *Italic* below.

Responses to reviewer #1

Major comments

The introduction lacks the typical emission ratios of $CH_4$, $N_2O$, $NH_3$, etc, in the agricultural fields. The authors use several cylinders to release these gases at known fluxes, but are these fluxes reasonable compared to the real cases?

*Emission ratios of $CH_4$, $N_2O$, $NH_3$ from agricultural fields can be varied. The release rates were point sources, not distributed as cattle or soil would be. The aim is to show that the known fluxes can be retrieved from the measurements, for all three gases. In this case it is permissible to have higher emissions than those typical in the field to minimise uncertainty due to background variability. We have added this in the revised manuscript on page 7, Line 140-142.*

It would be very useful to add one section to use the measured VMR from two OP systems to calculate the emission rates. The potential readers are more interested in the fluxes calculation instead of VMR measurements.

*Agree with reviewer's comment. we have added a section of flux calculation using backward Lagrangian stochastic dispersion modelling (WindTrax) and compared it to the known release rate (recovery study). Please see the section 3.5, 3.6 herd of cattle emission study and 3.7 Windtrax sensitivity in the revised manuscript, page 22-24, the figures A1-A7 and Table A1 are attached at Appendices. We also added a sentence in the Abstract "The mole fractions measured by OP-FTIR and OPL were also input into models of atmospheric dispersion (WindTrax) allowing the calculation of fluxes. Trace gas release recoveries with Windtrax were examined by comparing the ratio of calculated and known fluxes."*

Minor comments:

Line 31: $NH_3$ is an important atmospheric gas, but it is not GHG
*Yes, agree with the reviewer. It is worth to mention in the manuscript that ammonia ($NH_3$) is an indirect greenhouse gas, because $NH_3$ deposition can ultimately increase landscape emissions of nitrous oxide ($N_2O$), a potent greenhouse gas (GHG). These "secondary offsite" emissions, generally referred to as indirect emissions, are an important component of the GHG budget of the agricultural sector. The change can be seen in the revised manuscript, page 1, Line 32-33.*

Line42: "relies on" -> reply on.
*Yes, agree with the reviewer. The change has been made to be "reply on", see page 2, Line 46.*

Line 99: add information on why there are two pathlengths for some experiments.
*Yes, we added more information to explain the different path length. "The different path length was determined depending on the factors of wind conditions (direction and wind speed) and the distance between the path length and source area. Given the constant wind direction, the longer pathlength was needed when the measurement path was further away from the source so that the gas plume could pass by most of the OP measurement path." See page 8, Line 152-155.*

Line 107: "and 1 atm pressure" -> and 1 atm.
*Yes, agree with the reviewer. "the pressure" has been removed in the revised manuscript. See page 5, Line 113.*

Line 120: "the measurement path was 137 and 125 m (two-way path) for path 1 and 2, respectively." - > the measurement pathlengths were 137 and 125 m (two-way path) for paths 1 and 2, respectively.
*Yes, agree with the reviewer. The change is made to be "above ground level" in the revised manuscript, page 6, Line 126-128.*

Line 136: "wind conditions were such poor". Do you have the wind information? The poor wind means the wind direction is opposite? Or the wind speed is too fast?
*Yes, we agree with the reviewer. We added the information of poor wind conditions (E winds dominated) in the revised manuscript, page 7, Line 145.*

Line 120. "above ground level". Other places use above the ground, please be consistent throughout the whole manuscript.
*Yes, changes have been made in the revised manuscript.*

Line 145-146 (The layout of the experiment is not shown here): Please add it too.
*Yes, we agree with reviewer's comment. In the revised manuscript, we have added the layout of gas release trial at Wollongong on 3 August 2005 (Figure 3), as well the layout of two OP-FTIR comparisons at a commercial feedlot in Victoria in February 2008 (Figure 4). See pages 8, 9.*

[Figure]

*Figure 3: Point gas release sources and OP-FTIR path geometries (distances in m) at Wollongong August 2005. The FTIR measurement path lengths at day 1 and 2 were 87.5 and 150 m (two-way path), respectively. Three ¼" tubes coming from three tanks ($CH_4$ (nature gas), $NH_3$ and $N_2O$) bundled together on a stake at the release height 1.28 m above ground level.*

[Figure]

*Figure 4: Two OP-FTIR (Bomem MB100 and Bruker) during side-by-side operation in a commercial feedlot in Victoria in February 2008.*

Line 156. First time mentioning CO2 here. It might be better to add some introduction in Section 1.

*As suggested by Reviewer 2, $CO_2$ is not relevant to this study, and we agreed to remove the section that was associated with $CO_2$. Please see the change in the revised manuscript.*

Line 210: First time mentioning CO here. Same as CO2, it is good to add more information in Section 1.

*Same as above response, we have removed CO section in the revised manuscript.*

Line 209-210: the three spectral ranges are recorded simultaneously or individually with specific optical filters?

*The FTIR spectrometer measures the broadband IR spectrum simultaneously over the range 600-5000 $cm^{-1}$. The three separate spectral regions ($N_2O$ (2130–2283 $cm^{-1}$), $CH_4$ (2920–3020 $cm^{-1}$), and $NH_3$ (900–980 $cm^{-1}$)) are extracted from the broadband spectrum and analysed separately for each target species. Please see the changes in the revised manuscript page 10-11, Line 230-232.*

Line 234 : "14:45- 16:30" the time is local time or UTC?

*The time of measurement period was local time. We added "(local time)" in the revised manuscript, please see page 12, Line 282-283.*

Line 235: "From 14:45-15:10" -> Between 14:45 and 15:10.

*Yes, agree with the reviewer. We made the change in the revised manuscript, please see page 12, Line 283.*

Line 310: (data not shown). Please show it.

*We added the figure of comparing $NH_3$ mole fractions between two lasers and OP-FTIR (Fig. 9) on page 21.*

[Figure]

*Figure 9: 30-minute averaged NH₃ mole fraction measured by OP-FTIR and OPL unit (1015) positioned side-by-side (path length = 148 m) at Wollongong site.*

Line 322: what is "in stable conditions"?

*Monin-Obukov length L, L≅0–10 m. we added it in the revised manuscript, page 22, Line 415.*

Line 323: "Carbon dioxide results from both FTIRs were lower than those of air samples by approximately 15%." Do you have a reason for that?

*We have removed this statement as $CO_2$ is out of this study.*

Responses to reviewer #2

Reviewer suggested to add a discussion on uncertainty budget and calculate emission rates as a real-case scenarios.

*We agree with the reviewer. The initial manuscript described the measurements of side-by-side FTIR-laser comparisons and demonstrated excellent correlations between measured line-averaged mole fractions. Ratios of mole fractions in most cases were equal to ratios of release rates within 2%. The measurements confirmed the underlying principle that the released gases were equally dispersed by the atmosphere between source and measurement path. Scatter in measured ratios was larger than the measurement precision would imply and suggests that the random nature of atmospheric dispersion is the limiting factor to accuracy in the overall open path techniques, not instrumental measurement precision.*

*In the revised manuscript, following reviewer's suggestion, we added three sections:*

*1, the trial release experiments at Kyabram in July-August 2005 were also analysed using WindTrax to calculate gas source strengths from measured line average mole fractions for each gas. For uniform area sources, excellent recoveries were consistently obtained for $N_2O$ with Windtrax-calculated mole fractions in agreement with measured mole fractions within 5%. For $CH_4$ the results were often affected by additional local sources of $CH_4$ (local dairy herd) and for $NH_3$ there was evidence of losses at the ground surface leading to reduced $NH_3$ recoveries.*

*2, We also added a study on measuring GHG emissions from a herd of grazing cattle (353 cattle) using side-by-side OP-FTIR and OPL with sonic anemometer data and WindTrax for dispersion analysis. This experiment assessed the usefulness of the WindTrax approach when the source is not strictly uniformly distributed (grazing cattle for $CH_4$, scattered dung and urine patches for $NH_3$). Realistic results were obtained (average 290 g cow$^{-1}$ day$^{-1}$ for $CH_4$, and 0.3-0.8 $\mu g$ $m^{-2}$ $s^{-1}$ for $NH_3$), which was comparable to the results from our previous study. The $CH_4$ laser results for this trial are in good agreement with the FTIR results.*

*3, we added the uncertainty budget:*

*We want to compute the total uncertainty associated with the difference in mole fraction between upwind and downwind. There are three uncertainty sources: instrument precision uncertainty, fitting uncertainty, and absorption cross-section (HITRAN) uncertainty (the latter two are fractional uncertainties and were taken from Paton-Walsh et al. 2014). The measurement precision is in units of ppbv and so the fractional uncertainty that this represents will change with the trace gas mole fraction. The instrument precision uncertainty ($\delta$) associated with upwind measurement is 1-$\sigma$, and the uncertainty associated with downwind is also 1-$\sigma$. We assume these errors to be independent. The instrument precision uncertainty in the difference in mole fraction between upwind and downwind is thus $sqrt((1-\sigma)^2 + (1-\sigma)^2)$. We then divide this value by the difference in mole fraction to recover the relative uncertainty due to instrument precision: $sqrt((1-\sigma)^2 + (1-\sigma)^2) / (CH4_{downwind} - CH4_{upwind})$. $\Delta CH_4 = CH4_{downwind} - CH4_{upwind}$. We then add in quadrature the relative measurement uncertainty due to instrument precision with the fitting and absorption cross-section uncertainties (also expressed in terms of relative uncertainty). The unit for precision (1-$\sigma$) is ppbv, and % for uncertainty.*

*For example, for $CH_4$, when $\Delta CH_4$ was as low as 20 ppbv, we have a relative uncertainty of 0.28 for the instrument precision, 0.02 for fitting uncertainty, and 0.05 for absorption cross-section uncertainty. The relative uncertainty propagated across these three components is: $sqrt(0.28^2 + 0.02^2 + 0.05^2) = 0.283$ or 28.3%. When the $\Delta CH_4$ was increased to 50 ppbv or 100 ppbv, the uncertainty declined dramatically to 11.3 and 5.7%, respectively. However, for $N_2O$ and $NH_3$ the uncertainty was not limited by the mole fraction enhancement but likely attributed to absorption cross-section uncertainty.*

| | CH4 | N2O | NH3 |
|---|---|---|---|
| | $CH_4$ | $N_2O$ | $NH_3$ |
| Measurement precision (ppbv) | 4 | 0.3 | 0.4 |
| Spectral fitting uncertainty (%) | 2% | 4% | 2% |
| Absorption cross-section uncertainty (%) | 5% | 5% | 5% |
| δ(Δ trace gas mixing ratio)/Δ trace mole fraction (%) | | | |
| Δ trace gas mole fraction (ppbv) | | | |
| 20 | 28.3% | 2.1% | 2.8% |
| 50 | 11.3% | 0.8% | 1.1% |
| 100 | 5.7% | 0.4% | 0.6% |
| Total uncertainty (%) | | | |
| Δ trace gas mole fraction (ppbv) | | | |
| 20 | 28.8% | 6.8% | 6.1% |
| 50 | 12.5% | 6.5% | 5.5% |
| 100 | 7.8% | 6.4% | 5.4% |

Minor comments:

1) please show the layout of the experiments at three experimental sights.

*Agree with the reviewer. we have added the layout of the experimental site at Wollongong (Fig. 3) and commercial feedlot (Fig. 4).*

[Figure]

*Figure 3: Point gas release sources and OP-FTIR path geometries (distances in m) at Wollongong August 2005. The OP-FTIR measurement path lengths at day 1 and 2 were 87.5 and 150 m (two-way path), respectively. Three ¼" tubes coming from three tanks (CH₄ (nature gas), NH₃ and N₂O) bundled together on a stake at the release height 1.28 m above ground level.*

[Figure]

*Figure 4: Two OP-FTIR (Bomem MB100 and Bruker) during side-by-side operation in a commercial feedlot in Victoria in February 2008.*

2) measurement data of CO and $CO_2$ is presented in e.g. 2.4.2, however, it is out of the scope of the paper. Please remove or elaborate more.

*Agree. we have removed the section about CO and $CO_2$ in the revised manuscript.*

3) L272, not shown data is important for the discussion on the instrument precision, please include in the main text.

*Agree. we have added Figure 6 in the revised manuscript.*

[Figure]

*Figure 6: Regression/correlation analysis of the OP-FTIR measured enrichments from 14:45 to 16:25 of NH₃ vs N₂O (A) and CH₄ vs N₂O (B).*

4) L310 (data not shown), data can be included in the supplement.
*Agree. we have added Figure 9 in the revised manuscript, please see page 21.*

[Figure]

**Figure 9: 30-minute averaged NH₃ mixing ratio measured by OP-FTIR and OPL unit (1015) positioned side-by-side (path length = 148 m) at Wollongong site.**

5) Figures 3-5, please include the Y - error bars.

*We added standard errors (s.e.) to the 5-min averages of CH₄ and NH₃ mole fraction measured at Kyabram (Figure 7) and s.e. to the 30-min mean of CH₄ mole fraction measured by OP-FTIR and OPL at Wollongong (Fig. 8). However, we did not add error bars in Figure 5 as each dot represent a single measurement (i.e., raw data) which makes it impossible to compute the standard error associated with each dot.*

[Figure]

***Figure 7: 5-minute averages of CH₄ (A) and NH₃ (B) mole fraction measurements from the OP-FTIR and OPL downwind of a ground-level grid source 40 × 15 m wide (path length = 125 m) at Kyabram on 3 August 2005 (T2).***

[Figure]

*Figure 8: Thirty-minute averages of CH₄ mole fraction measured by OP-FTIR and both OPL units (1012 and 1013) positioned side-by-side (path length = 148 m) at Wollongong site. Error bars denote the standard error.*